# WatermarkLab: A Comprehensive Framework for Robust Image Watermarks Benchmarking and Development

## Abstract

With the growing demand for multimedia content, image protection has become increasingly important. Robust image watermarking, as a core technology for copyright protection, has attracted extensive attention. To advance research in this field, we propose WatermarkLab, a comprehensive framework for systematic benchmarking of robust image watermarks and the development of new methods. WatermarkLab supports benchmarking of all types of *blind* robust image watermarks, including *in-generation* watermarks and *post-generation* watermarks. Beyond benchmarking, WatermarkLab integrates 10 representative watermarking methods for systematic comparison. It also includes 34 attackers for benchmarking and 28 differentiable attackers for development. Furthermore, we evaluate the robustness of 9 watermarking methods under 34 attackers and give their weaknesses, assisting researchers in enhancing more robust watermarking methods and designing new watermark removal attackers. In addition, the framework provides auxiliary tools such as arithmetic coding and *reversible data hiding* commonly used in *robust reversible watermarking*. For result visualization, WatermarkLab offers comprehensive visualization tools and an interactive website, enabling researchers to intuitively analyze and compare benchmarking results. In summary, WatermarkLab is a powerful framework, aiming to establish a *comprehensive*, *fair*, *open*, and *extensive* platform for *blind robust image watermark* benchmarking and development. Interactive visualization website code is available at: https://anonymous.4open.science/r/watermarklab-website.

## 1 Introduction

With the rapid development of digital media, image content protection has become a critical area of research. Unauthorized reproduction, modification, and distribution are particularly prominent in industries such as photography, art, and media. The rise of generative models, such as Stable Diffusion Rombach et al. (2022), DALL·E Ramesh et al. (2022) and Midjourney Midjourney (2022), has further intensified this challenge by enabling large-scale generation of highly realistic synthetic content Zhao et al. (2025); Wang et al. (2024). Artificial Intelligence-Generated Content (AIGC) introduces unprecedented challenges to copyright protection and content authentication. Governments around the world have paid great attention to this issue and actively promote watermarking technology as a key means to ensure content provenance and to combat disinformation European Parliament and Council (2024); Executive Office of the President (2023); Cyberspace Administration of China (2025). For this reason, robust image watermarking, which can effectively achieve content provenance and authentication without significantly degrading visual quality, has attracted wide research attention Tancik et al. (2020); Wen et al. (2023); Fernandez et al. (2023); Yang et al. (2024); Gunn et al. (2024); Lu et al. (2024); Xu et al. (2025); Bui et al. (2023a).

Robust image watermarking ensures traceability and content integrity authentication by embedding perturbations that carry identification information into images. These perturbations act as unique markers that can still be extracted after unauthorized distribution. We classify watermarking into two types according to the embedding stage. 1) **In-Generation Watermarking (IGW)**, as shown in Figure 1(a). In this type, watermarks are embedded during the image generation process. Generative models directly add watermark signals, often as latent variables that are mapped into protected

Figure 1: Two types of image watermarking: (a) **In-Generation Watermarking (IGW)**, embedding watermarks during image generation; (b) **Post-Generation Watermarking (PGW)**, embedding watermarks into pre-existing images after generation.

images. 2) **Post-Generation Watermarking (PGW)**, as shown in Figure 1(b). In this type, watermarks are embedded into pre-existing cover images after generation. This is usually done through an encoder, a noise layer, and a decoder in the training pipeline. After embedding, protected images may go through lossless or lossy channels before extraction. Since protected images may undergo various distortions during transmission, systematic robustness evaluation is crucial for the development of reliable watermarking methods.

To address this need, we propose WATERMARKLAB, a comprehensive and extensible framework for evaluating and developing both *PGW* and *IGW* methods. Our main **contributions** are as follows:

- WATERMARKLAB supports the benchmarking and development of all types of *blind* robust image watermarking algorithms, including *zero-bit* and *multi-bit* watermarking, *PGWs*, *IGWs*, as well as *robust reversible watermarking* that can detect arbitrary attacks and recover the original cover image losslessly. To facilitate customized benchmarking, WATERMARKLAB integrates 10 representative methods, such as *PGW* methods: DctDwt Cox et al. (2007), DctDwtSvd Cox et al. (2007), RivaGAN Zhang et al. (2019), StegaStamp Tancik et al. (2020), TrustMark Bui et al. (2023a), InvisMark Xu et al. (2025), and the VINE Lu et al. (2024); *IGW* methods: Tree-Ring Wen et al. (2023), StableSignature Fernandez et al. (2023), GaussianShading Yang et al. (2024). Researchers can directly load these models for benchmarking or use them as baseline methods for further study. In addition, WATERMARKLAB adopts a modular design, allowing researchers to extend `base` classes to benchmark their own watermarking methods.

- For benchmarking, WATERMARKLAB incorporates six categories of attack methods, including compression, color transformation, geometric distortion, noise addition, diffusion-based regeneration, and filtering, totaling 34 attackers. In addition, it provides 28 commonly used differentiable attackers, which can be directly applied in adversarial training to help researchers develop more robust watermarking methods. The framework also includes auxiliary tools, such as compression coding Witten et al. (1987) and *reversible data hiding* Ni et al. (2006), to support the development of *robust reversible watermarking* methods Chen et al. (2025). Its modular design ensures that all metrics, datasets, and watermarking methods can be customized and seamlessly integrated into evaluations.

- We evaluated 9 watermarking schemes, including 3 *IGWs* and 6 *PGWs*, and analyzed their robustness under 34 attackers. The results show that while mainstream IGWs perform well under most attacks, they still exhibit limitations in certain cases. Further comparative evaluation of *PGW* and *IGW* methods indicates that StegaStamp remains the benchmark with the strongest robustness among *PGWs*, whereas GaussianShading leads the *IGWs* but still shows some vulnerability to geometric distortions. To facilitate analysis, WATERMARKLAB provides a set of built-in visualization tools and an interactive website (https://anonymous.4open.science/r/watermarklab-website), allowing researchers to upload their results and perform intuitive comparisons with existing methods.

## 2 RELATED WORK

**Robust image watermarking** can be classified into two main types based on the embedding stage: *IGWs* and *PGWs*. *IGWs* watermark bits simultaneously with image generation, ensuring seamless integration during the process, whereas *PGWs* applies watermark embedding separately after the image has been fully generated.

**IGWs** primarily protect images generated by *text-to-image generative models*, such as Stable Diffusion Rombach et al. (2022), DALL·E Ramesh et al. (2022), and Midjourney Midjourney (2022).

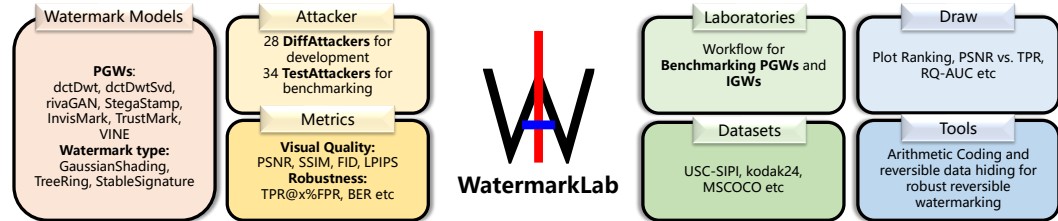

Figure 2: Architecture of WATERMARKLAB, which adopts a modular design to flexibly support customized benchmarking and new robust image watermarking development.

In these approaches, watermark bits typically act as latent variables, often combined with textual prompts, and are carefully mapped into a protected image through a generator and watermark encoder. Representative methods include RoSteALS Bui et al. (2023b), Tree-Ring Wen et al. (2023), StableSignature Fernandez et al. (2023), and GaussianShading Yang et al. (2024), PRC Gunn et al. (2024). Tree-Ring embeds the watermark, represented by structured patterns, into the initial latent variable and maps it together with the prompt into a protected image using diffusion models Ho et al. (2020); Song et al. (2021). During extraction, the attacked protected image is mapped back to the latent variable containing the watermark via DDIM inversion Song et al. (2021), and the watermark are then detected. RoSteALS embeds watermark bits into generated images by injecting them into the latent codes of frozen pretrained autoencoders. This approach preserves image quality, ensures robust secret recovery under distortions, and can be adapted to cover-less or text-conditioned watermark. StableSignature fine-tunes the VAE decoder of a latent diffusion model Rombach et al. (2022) to embed an invisible watermark conditioned on a binary signature. A pre-trained decoder recovers the hidden signature from generated images, and a statistical test verifies its origin with high accuracy under diverse complex conditions. GaussianShading embeds the watermark by mapping watermark bits into an initialized latent variable following a Gaussian distribution. Similar to Tree-Ring, the watermark is extracted from the attacked image using DDIM inversion. The PRC watermark is an undetectable method that embeds watermarks using a pseudorandom error-correcting code, in a manner similar to GaussianShading, preserving image quality while ensuring robustness.

**PGWs** can be categorized into two types based on the embedding approach: *non-learning-based watermarking* and *learning-based watermarking*. Non-learning-based methods rely on traditional signal processing techniques to embed watermark bits into a cover image. Learning-based methods, on the other hand, leverage machine learning frameworks—typically following an *encoder-differentiable attacker-decoder* architecture—to learn watermark embedding and extraction. Representative non-learning-based methods include DctDwt Cox et al. (2007) and DctDwtSvd Cox et al. (2007). Learning-based watermarking methods often incorporate a differentiable attacker to enhance robustness against various distortions. Notable examples include HiDDeN Zhu et al. (2018), TrustMark Bui et al. (2023a), InvisMark Xu et al. (2025), and VINE Lu et al. (2024). Some methods are specifically designed to resist physical distortions: for instance, StegaStamp Tancik et al. (2020) demonstrates high reliability under print-capture attacks, while LFM Wengrowski & Dana (2019) and PIMoG Fang et al. (2022) exhibit resilience against screen-capture attacks in scenarios like photography or display. Additionally, the *robust reversible watermark* CRMark Chen et al. (2025) can achieve lossless recovery of the cover image in noiseless channels while maintaining robust extraction in lossy channels. In summary, the main advantage of these *PGWs* is their ability to be trained end-to-end using differentiable attackers, yielding strong robustness under diverse attackers.

**Benchmarking for Watermarking.** Recent efforts have sought to establish standardized benchmarks for evaluating the robustness of watermarking methods. WAVES An et al. (2024) is the first comprehensive platform that systematically evaluates deep learning watermarking techniques against generative model-driven manipulations. It covers a broad range of distortion types, including compression, regeneration Zhao et al. (2024), embedding attack Dong et al. (2023), and surrogate detector attacks Saberi et al. (2023), as well as common image processing operations. However, its evaluation scope is limited to three representative watermarking methods: StegaStamp Tancik et al. (2020), Tree-Ring Wen et al. (2023), and Stable Signature Fernandez et al. (2023). More recently, W-Bench Lu et al. (2024) has been proposed to focus on robustness evaluation under image editing scenarios, emphasizing its support for watermarking models applicable to arbitrary images. Despite these advances, a truly *open, fair, comprehensive, and extensible* framework for systematic benchmarking and development platform of robust watermarking is still required.

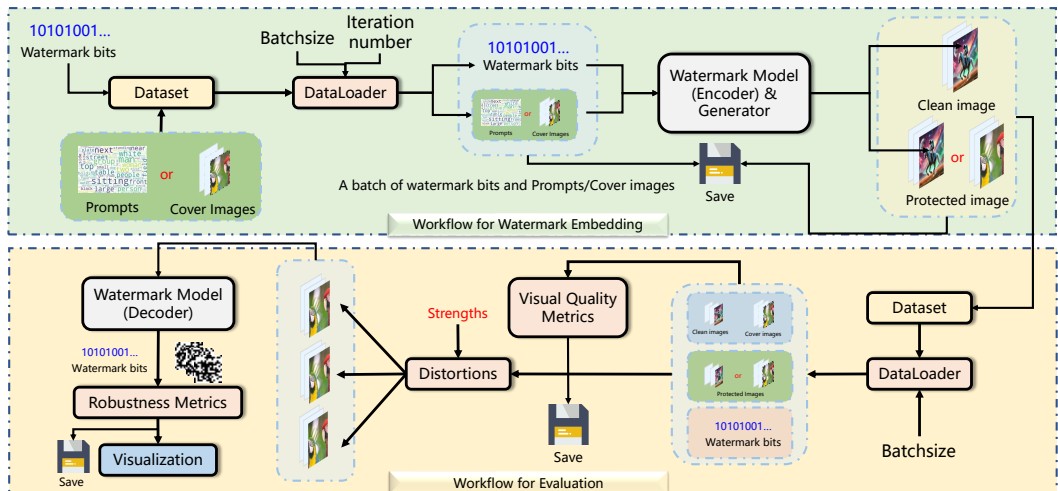

Figure 3: Workflow of WATERMARKLAB, illustrating the two main stages: embedding and evaluation. The workflow support both *PGW* and *IGW* methods.

## 3 WATERMARKLAB

WATERMARKLAB adopts a modular and extensible architecture for flexible benchmarking and development of robust image watermarking techniques. As shown in Figure 2, it consists of seven core modules. The `watermarks` module integrates 10 representative *PGW* and *IGW* methods for comparative studies. The `attackers` module includes differentiable and testing attackers for adversarial training and evaluation. The `metrics` module provides robustness and imperceptibility metrics for systematic assessment. The `tools` module offers utilities such as compression codecs and reversible data hiding. The `datasets` module integrates mainstream datasets for benchmarking. The `laboratories` module serves as an evaluation platform for *PGW* and *IGW* methods. Finally, the `draw` module enables visualization and interactive analysis of results.

As shown in Figure 3, the workflow of WATERMARKLAB is divided into two main phases: embedding and evaluation. The embedding phase integrates watermarks into cover using modular strategies, supporting both *IGWs* and *PGWs*. The evaluation phase primarily tests the robustness of watermarks under various distortions and the visual quality of protected images.

**Workflow for Watermark Embedding:** To support both *IGWs* and *PGWs* evaluation, WATERMARKLAB employs a modular design just like PyTorch Paszke et al. (2019). Users can implement the `dataset` interface to load custom datasets, allowing evaluation across various input formats such as text prompts or cover images. `DataLoader` enables efficient batch processing. To handle random distortions (e.g., Gaussian noise, Salt&Pepper noise), an iteration parameter allows embedding different watermark bits into the same cover image for statistically robust evaluation. Researchers can also implement the watermark model interface to test custom watermarking methods.

**Workflow for Evaluation:** Following embedding, WATERMARKLAB assesses the visual quality and robustness of protected images. The images are subjected to various attacks, and the watermark is extracted using a user-defined decoder to calculate extraction accuracy. All results, including images, extracted watermark bits, and evaluation metrics, are saved and can be visualized via the built-in `draw` module, which generates comparison curves. Most importantly, evaluation results can be uploaded to the interactive visualization platform.

**Datasets for Evaluation:** WATERMARKLAB incorporates a wide range of widely used datasets, such as the MS-COCO 2017 Lin et al. (2014) image set and a caption collection for *IGW* evaluation, as well as classical benchmarks including Kodak24 Eastman Kodak Company (1999) and USC-SIPI USC, SIPI (1977). Beyond these, WATERMARKLAB adopts a modular design that enables researchers to seamlessly integrate their own datasets. This extensibility also allows for benchmarking on emerging generative datasets, such as Diffusion-DB Wang et al. (2022) and DALL·E3.

**Metrics to Robustness and Visual Quality Evaluation:** To comprehensively evaluate the effectiveness of watermarking methods, WATERMARKLAB provides a set of metrics for both robustness and visual quality assessment. ***Robustness Metrics*** are primarily used to evaluate the ability of

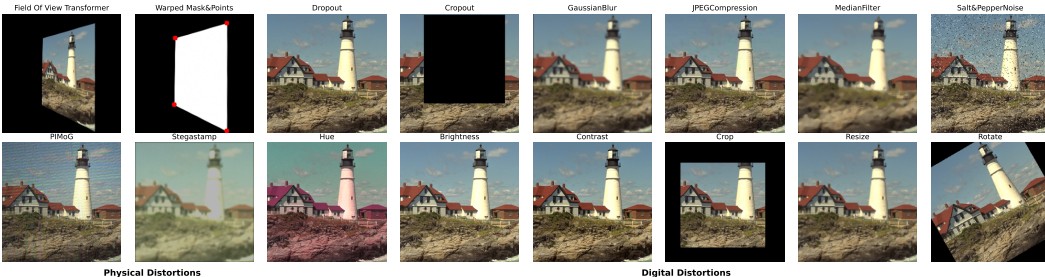

Figure 4: Illustration of 16 representative *DiffAttacker* examples supported in WATERMARKLAB, which can be used for *PGWs* development

watermarks to withstand various attacks. WATERMARKLAB integrates a set of commonly used robustness evaluation metrics, including the Bit Error Rate (BER), which quantifies the proportion of incorrectly extracted bits relative to the total embedded bits, and Extraction Accuracy (EA), which measures the rate at which protected images are correctly identified under different distortion conditions. Additionally, the true positive rate at $x\%$ false positive rate (TPR@$x\%$FPR) is employed for watermark detection. This metric applies not only to *multi-bit watermarks* but also to *zero-bit watermarks*, and has been widely adopted by methods such as Tree-Ring Wen et al. (2023), StableSignature Fernandez et al. (2023), and GaussianShading Yang et al. (2024) for assessing watermark existence and robustness. *Visual Quality Metrics* are used to evaluate the impact of watermark embedding on the perceptual quality of images. WATERMARKLAB incorporates widely adopted metrics, including Peak Signal-to-Noise Ratio (PSNR), Structural Similarity Index Measure (SSIM) Wang et al. (2004), Learned Perceptual Image Patch Similarity (LPIPS) Zhang et al. (2018), and Fréchet Inception Distance (FID) Heusel et al. (2017). These metrics enable an assessment of image distortions and perceptual similarity from signal-level, structural, and perceptual perspectives, assisting researchers in balancing robustness and imperceptibility. Moreover, WATERMARKLAB supports user-defined metrics by providing a base class for inheriting and implementing custom evaluation methods.

**Attackers for Training and Testing:** WATERMARKLAB provides two types of attackers for watermark development and benchmarking. *TestAttacker* includes 34 different attacks, which can be grouped into several categories: **1) Color Transformations** Smith (1978): saturation, brightness, contrast adjustment, and color quantization; **2) Noise Attacks:** Gaussian noise, Poisson noise Foi et al. (2008), and salt-and-pepper noise Pratt (2007); **3) Non-Learned-based Compression and Learned-based Compression**: JPEG Wallace (1991), JPEG2000 Skodras et al. (2002), WebP compression Google (2023), end-to-end and VAE-based compression Ballé et al. (2016), Ballé et al. (2018); Minnen et al. (2018); Cheng et al. (2020); **4) Filter Attacks**: Gaussian Blur, Mean Filter etc; **5) Diffusion Model-based Regeneration** Zhao et al. (2024). **6) Geometric Transformations**: scaling, rotation etc. Each attacker type can be applied at multiple attacking strengths to evaluate the robustness of watermarking under different strengths. *DiffAttacker* serves as differentiable attacker for end-to-end training, introducing learnable attacks that allow the model to adaptively enhance its resistance to these attacks. *DiffAttacker* mainly inherits 28 types of attackers, partial attacker shown in Figure 4, including differentiable JPEG compression (JPEG-Polynomial Tancik et al. (2020), JPEG-Fourier Xing et al. (2021), and JPEG-Mask Zhu et al. (2018)), Gaussian noise injection, color transformations, geometric transformations, as well as physical distortions simulating screen-capture (PIMoG Fang et al. (2022)) and print-capture (StegaStamp Tancik et al. (2020)). Overall, *DiffAttacker* provide an effective means to enhance robustness during training, supporting the development of new watermarking methods. Moreover, both *TestAttacker* and *DiffAttacker* support custom extensions, enabling evaluation of new attacks and the development of novel watermarking techniques.

## 4 EVALUATION

We evaluate two categories of watermarking methods: *IGW* and *PGW*. The IGWs include Tree-Ring Wen et al. (2023), StableSignature Fernandez et al. (2023), and GaussianShading Yang et al. (2024), where the diffusion model used for the *IGWs* is `StableDiffusion v2.1-Base`, while the *PGWs* consist of dctDwtSvd Cox et al. (2007), RivaGAN Zhang et al. (2019), Stegatamp Tancik et al. (2020), Trustmark Bui et al. (2023a), VINE-R Lu et al. (2024), and Invismark Xu et al. (2025).

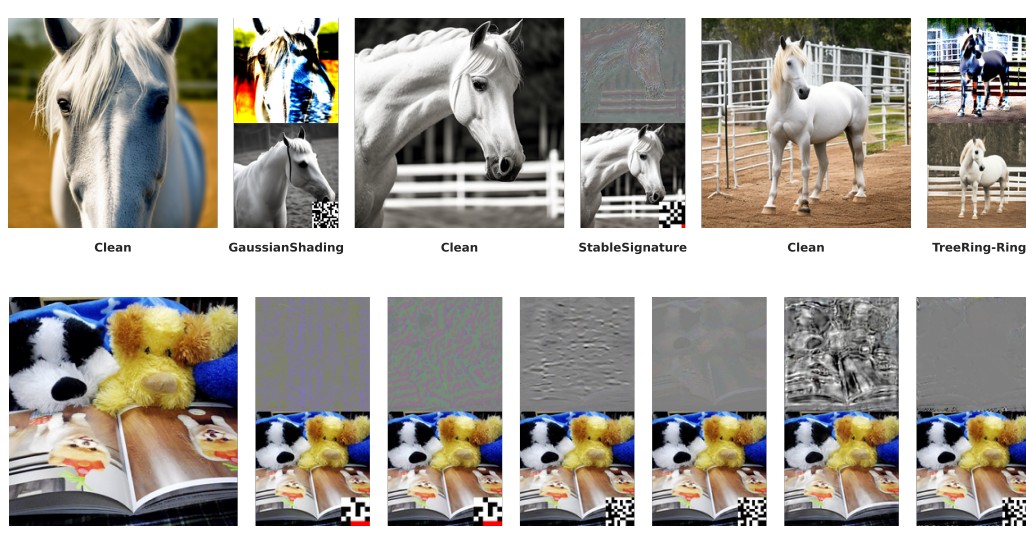

Figure 5: Visualization comparison of cover/clean images, protected images, and residual images for three *IGW* methods (First row: Tree-Ring Wen et al. (2023), StableSignature Fernandez et al. (2023), GaussianShading Yang et al. (2024)) and six *PGW* methods (Second row: dctDwtSvd Cox et al. (2007), RivaGAN Zhang et al. (2019), Trustmark Bui et al. (2023a), and Invismark Xu et al. (2025), Stegatamp Tancik et al. (2020), VINE-R Lu et al. (2024)).

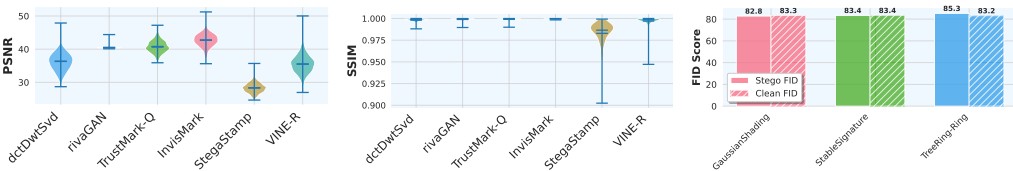

Figure 6: Quantitative comparison of visual quality using PSNR, SSIM, and FID metrics.

The evaluation primarily focuses on robustness and visual quality, and further analyzes the effectiveness of attackers. The primary dataset used is MS-COCO 2017 Lin et al. (2014), and robustness is systematically evaluated under 34 attackers described in Table 1 (Appendix C).

**Benchmarking The Visual Quality:** Figure 5 visualizes the cover/clean images, protected images, and corresponding residual image for three *IGWs* and six *PGWs*. For *IGWs*, it can be observed that Tree-Ring Wen et al. (2023) exhibits noticeable differences between the clean and protected images. This is primarily because Tree-Ring embeds watermarks by modifying the initial latent variables, which alters the distribution of the initial latent variables and consequently changes the generated image content. In contrast, StableSignature Fernandez et al. (2023) embeds watermarks by fine-tuning the decoder of the VAE in the *Latent Diffusion Model* Rombach et al. (2022), resulting in smaller visual differences between the clean and protected images. This approach is visually more similar to *PGWs*. GaussianShading Yang et al. (2024) differs from the previous two methods by directly mapping *multi-bit watermarks* to the initial latent variables. Due to the mapped initial latent variables follows a Gaussian distribution, therefore, Yang et al. (2024) state that Gaussian-Shading is "lossless performance" robust image watermarking. Therefore, compared with TreeRing and StableSignature, GaussianShading achieves a dual improvement in lossless performance and higher payload. Its only drawback is that it reduce the diversity of generated images Zhao et al. (2025). For *PGWs*, all five methods exhibit high imperceptibility; However, residual images show that RivaGAN and DctDwtSvd introduce noticeable color differences, indicating weaker coupling with image content. In addition, VINE-R Lu et al. (2024) introduces visible modification traces along image edges, which may attract the attention of potential attackers. Based on the quantitative metrics in Figure 6, the distributions of PSNR and SSIM indicate that TrustMark Bui et al. (2023a) and InvisMark Xu et al. (2025) achieve higher visual quality within the *PGWs*. For *IGWs*, FID met-

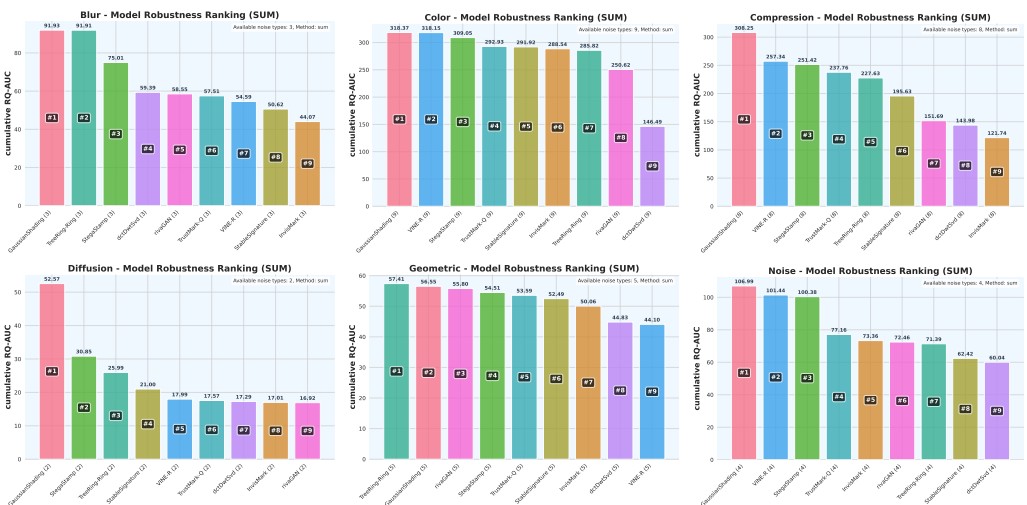

Figure 7: Model robustness ranking by *cumulative* **RQ-AUC** under the six types of 34 attackers.

rics reveal that GaussianShading achieves lower FID compared to StableSignature and Tree-Ring, further confirming that GaussianShading can maintain visually lossless performance.

**Benchmarking The Robustness:** We analyze the **RQ-AUC** scores of all models under 34 attacks, with results shown in Figure 18 in the Appendix C. This metric represents the area under the $\text{TPR}@x\%\text{FPR}$ vs. PSNR curve, where a larger value indicates stronger robustness (detailed definition is provided in Appendix A.2). For *PGWs*, almost all methods struggle against regeneration attacks, indicating the strong watermark removal ability of this category. Overall, StegaStamp Tancik et al. (2020) is more vulnerable to flipping and rotation attacks, while maintaining high robustness against other attacks. TrustMark Bui et al.

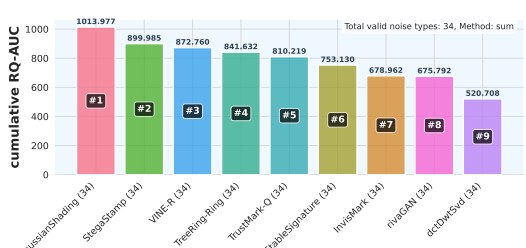

Figure 8: Overall ranking across all 34 attackers using *cumulative* **RQ-AUC**.

(2023a) ranks behind StegaStamp in robustness, exhibiting weaknesses against rotation, flipping, and regeneration attacks. By contrast, VINE Lu et al. (2024) not only shows vulnerabilities to regeneration, rotation, and flipping attacks, but is also highly sensitive to cropping. Although the original VINE paper reports robustness to regeneration, our results show a significant drop when using a different diffusion model for regeneration (`Stable Diffusion v2.1-base`), indicating that VINE has poor transferability and generalization against regeneration attacks. Moreover, We also find that VINE is highly sensitive to cropping. As shown in Figure 15 and 16, cropping just 10% of the image boundary sharply reduces watermark extraction accuracy. The TPR@0.1%FPR drops to 0.01. In contrast, under Cropout attacks, the watermark remains robustly extractable even when the crop ratio reaches 90%. We believe that VINE primarily embeds watermarks in boundary regions of the cover images, which explains the noticeable edge distortions observed in Figure 5. Therefore, its robustness against deep editing attacks likely stems from the watermark not being embedded in the main image content, making it difficult for deep editing to remove. Although introducing differentiable crop-based attacks could potentially improve VINE's robustness against cropping, it remains to be verified whether the improved VINE can still withstand deep editing attacks. For *IGWs*, similar to *PGWs*, they are particularly weak against regeneration and geometric distortions. For example, GaussianShading shows strong robustness against regeneration-based attacks but is relatively vulnerable to geometric attacks such as rotation and flipping. StableSignature and Tree-Ring exhibit similar vulnerability; however, Tree-Ring is slightly more robust than the other two under rotation attacks. This indicates that regeneration and geometric attacks are effective for *PGWs* and *IGWs*.

We evaluated the robustness rankings of all nine watermarking methods across different attack categories, as shown in Figure 7, using the **RQ-AUC** metric. It can be observed that StegaStamp and

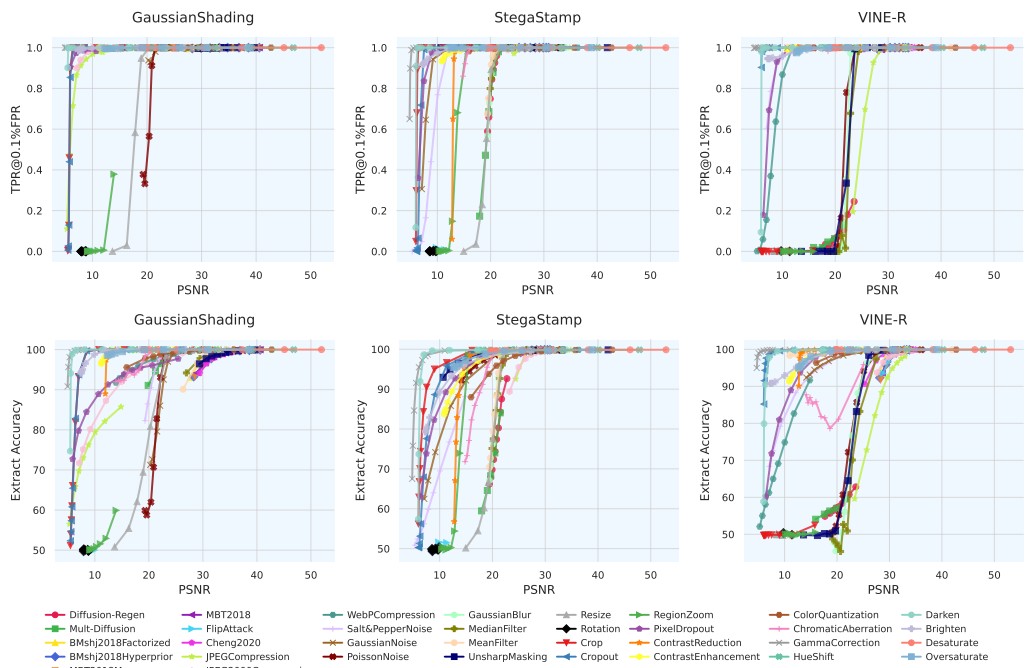

Figure 9: PSNR vs. TPR@0.1%FPR/Extract Accuracy curves of 3 watermark models (Gaussian-Shading Yang et al. (2024), StegaStamp Tancik et al. (2020), VINE Lu et al. (2024)) under 34 attacks. Curves closer to the bottom-right corner indicate stronger attacks, causing lower detection rates with less visual distortion. More results are provided in the Figure 19 and 20.

GaussianShading demonstrate strong robustness against most attackers, consistently ranking among the top across different distortion types. Furthermore, based on the **RQ-AUC** values, Gaussian-Shading and VINE exhibit relatively weak robustness against geometric attacks. This indicates that improving resistance to geometric attacks remains a key challenge for both GaussianShading and VINE. Figure 8 summarizes the overall rankings across all 34 attacks based on **RQ-AUC**. It can be observed that, regardless of the ranking strategy, GaussianShading wins the gold medal, StegaStamp the silver medal, and VINE the bronze medal. Although some *PGWs* were proposed after StegaStamp, StegaStamp remains the most robust watermarking method under most attacks. So far, GaussianShading and StegaStamp are still the most robust methods in the *IGW* and *PGW* categories, respectively, and remain the benchmarks that new methods need to surpass.

In the comparison between *PGW* and *IGW* methods, *IGW* methods clearly wins the gold medal, with a considerable lead over the second place. We attribute this mainly to the ability of *IGW* methods to directly map the watermark into the latent distribution of the generated image. For example, GaussianShading maps the watermark bits into a latent variable that follows a Gaussian distribution. The watermark is then diffused through the diffusion model into the entire image. This ensures that the watermark is distributed more evenly across the image and maintains a strong implicit correlation with the image content, resulting in stronger robustness. In contrast, *PGW* methods embed the watermark through an additional perturbation. In this case, the protected image and watermark are associated only through the perturbation, making *PGW* methods generally less robust than *IGWs*. However, compared with *IGWs*, *PGWs* also offer unique advantages. They can leverage end-to-end training to learn robustness against diverse attacks. For instance, StableSignature Fernandez et al. (2023) achieves strong robustness against RegionZoom distortions by fine-tuning a pretrained PGW decoder. Based on these insights, we believe that improving *IGW* robustness may require designing more reliable mappings between the watermark and the initial latent variables. Additionally, integrating such distributional mappings with end-to-end training could be a promising direction.

**Benchmarking The Attackers:** Figure 9 shows the PSNR vs. TPR@0.1%FPR curves of six models under 34 attacks to compare attack effectiveness. Curves near the bottom-right corner indicate stronger attacks, causing lower detection rates with less visual distortion. To better evaluate attack effectiveness, we use PSNR at a target TPR@$x$%FPR as the efficiency metric. For TPR@$x$%FPR =

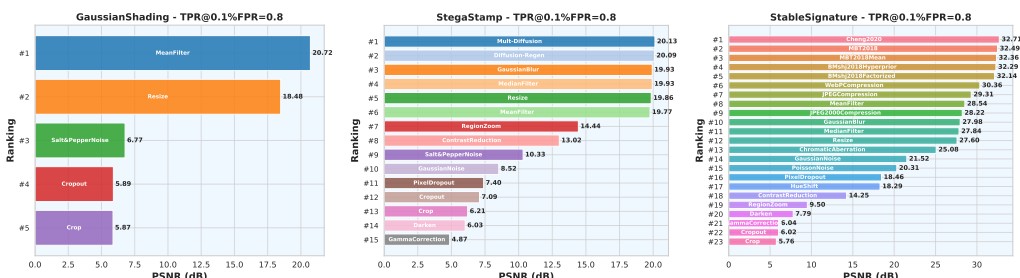

Figure 10: Attacker rankings reaching $r = 0.8$ for three models (GaussianShading, StegaStamp, and StableSignature), measured by PSNR@(TPR@0.1%FPR=0.8). Higher PSNR indicates greater attack efficiency. Results for all other models are provided in Figures 21–29.

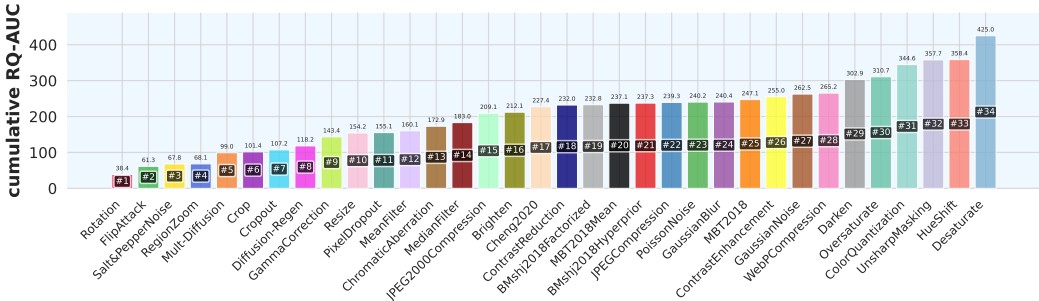

Figure 11: Attack strength ranking across 9 watermarking models based on the *cumulative* **RQ-AUC** metric, where lower values indicate stronger attacks.

$r$, a higher PSNR means a more successful attack with less visual damage. The metric is defined in detail in Appendix A.3. Figure 10 shows the rankings for $r = 0.8$. Note that some weaker attackers cannot reach the target $r$ and are therefore not shown in the figure. For GaussianShading Yang et al. (2024), only five attacks achieve the target $r = 0.8$, indicating that GaussianShading is significantly more robust than other methods. In terms of PSNR of the attacked images, MeanFilter achieves the highest attack efficiency. For StegaStamp and StableSignature, Multi-Diffusion Zhao et al. (2024) and Learning-based compression Ballé et al. (2016; 2018); Minnen et al. (2018); Cheng et al. (2020) achieve the best attack efficiency, respectively. Moreover, the PSNR values indicate that reducing TPR@0.1%FPR to 0.8 requires stronger image distortions for GaussianShading and StegaStamp, whereas StableSignature does not require such high-intensity attacks. This suggests that for robust watermarking methods, more efficient attackers are still needed; otherwise, attacked images will suffer unacceptable visual quality. Additionally, Figure 11 visualizes the ranking of attack strengths across different watermarking methods using the *cumulative* **RQ-AUC** metric, where lower values indicate stronger attacks. The results show that geometric and regeneration-based attacks occupy the top ranks, highlighting that most existing methods remain insufficiently robust against these two categories of attacks. It should be noted that this ranking only reflects the strength of attacks on the watermarking models and does not indicate the visual quality of the attacked images.

## CONCLUSION

We present WATERMARKLAB, a *comprehensive*, *fair*, *open*, and *extensible* benchmark and development framework for blind robust image watermarking. It integrates diverse methods, attacks, datasets, and evaluation metrics into a standardized platform for systematic robustness analysis and comprehensive evaluation. Extensive experiments on nine representative schemes under 34 attacks reveal the strengths and limitations of current *IGW* and *PGW* approaches, showing that StegaStamp and GaussianShading remain leading methods in *PGWs* and *IGWs*, respectively. The framework further provides built-in visualization tools and an interactive website for intuitive analysis, flexible benchmarking, and fair comparison of different watermarking approaches. We believe WATERMARKLAB will be a valuable framework, advancing digital image protection, supporting development of more robust methods, and collaborative research across the community.

## REPRODUCIBILITY STATEMENT

WATERMARKLAB aims to provide a *fair*, *open*, and *extensible* benchmark for robust image watermarking, with reproducibility as a key contribution. All experiments, figures, and results in this paper are generated using the WATERMARKLAB library and its evaluation and visualization modules. Integrated watermarking methods can be directly re-tested, and the interactive website offers APIs for reproducing all reported results.

## ETHICAL CONSIDERATIONS

In recent years, the misuse and malicious manipulation of AI-generated content have become increasingly prominent, posing potential ethical risks. Robust image watermarking, as an effective method for marking AI-generated content, can enhance the traceability and regulatory control of such content. The WATERMARKLAB platform proposed in this work aims to provide the community with a *fair*, *open*, and *extensible* benchmark and development environment for robust image watermarking, thereby promoting effective oversight of AI-generated content and generating positive societal impact. Given that WATERMARKLAB also includes certain watermark attack functionalities, to ensure research safety and responsibility, we emphasize that the platform should be used exclusively for scientific research and the development of protective mechanisms, and users must follow responsible usage guidelines.

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

## A PERFORMANCE METRICS FOR ROBUSTNESS AND ATTACK

### A.1 BLIND ROBUST IMAGE WATERMARKING

A digital watermarking system consists of three core modules: *embedding*, *extraction*, and *detection*. It can imperceptibly embed structured information into visual content and ensure that the information can be reliably recovered and detected even after lossy transmission or malicious tampering. This supports key applications such as content traceability, copyright verification, and generation-source authentication. In the *embedding* stage, the system takes a cover image $\mathbf{x}_o$ and a watermark message $\mathbf{m} \in \{0,1\}^L$ or $\mathbb{R}^L$, where $L$ denotes the payload capacity. Depending on the structure of $\mathbf{m}$, watermark methods can be divided into two types. If $\mathbf{m} \in \{0,1\}^L$ is a bitstream, it is called a *multi-bit watermark*. If $\mathbf{m} \in \mathbb{R}$ is a statistical signature with no explicit semantic meaning, it is called a *zero-bit watermark* and is only used for presence detection without conveying extra information. Methods such as Tree-Ring Wen et al. (2023) belong to this type. In addition, SSL watermarking Fernandez et al. (2022) also provides a solution for *zero-bit* watermarks. Based on this, the watermark embedding process can be formalized as a mapping function that injects the watermark into the cover image:

$$\mathbf{x} = \text{EMBED}(\mathbf{x}_o, \mathbf{m}), \tag{1}$$

where $\mathbf{x}$ is the protected image. From the perspective of rate-distortion theory, there is an inherent trade-off between the *robustness* and *imperceptibility* of a watermark. Therefore, watermark methods must carefully balance these two aspects to achieve an optimal solution, ensuring that the watermark remains both robust and visually unobtrusive. In the *extraction* stage, an estimate $\hat{\mathbf{m}}$ is recovered from an attacked protected image $\hat{\mathbf{x}} = \mathcal{A}(\mathbf{x})$:

$$\hat{\mathbf{m}} = \text{DECODE}(\hat{\mathbf{x}}) \tag{2}$$

This extraction process indicates that the system is *blind*, as it does not require any information from the original cover image. Blind watermarks have a major advantage over non-blind watermarks because they do not rely on any additional information for extraction. Furthermore, based on extraction results under a lossless channel, watermarks can be categorized into *robust image watermarks* and *robust reversible watermarks*. Compared to robust image watermarks, robust reversible watermarks can perfectly recover both the cover image and the watermark message in a lossless channel, i.e.,

$$\mathbf{m}, \mathbf{x} = \text{DECODE}(\mathbf{x}) \tag{3}$$

This special type of watermark can detect any attack and verify the integrity of the image content, such as CRMark Chen et al. (2025). Such methods not only ensure the authenticity and ownership of digital media but also provide strong guarantees for forensic analysis and content auditing.

### A.2 WATERMARK ROBUSTNESS METRICS

Images protected by watermarks are transmitted through various lossy channels and must support the extraction of watermark bits or the detection of watermark presence when needed. Consequently, robustness is a key criterion for evaluating the practical value of a watermarking system. Traditionally, robustness assessment focuses on quantifying the recovery of embedded bits, with common metrics including *bit accuracy* and *bit error rate*. These metrics measure robustness performance by comparing the original watermark $\mathbf{m}$ with the extracted result $\hat{\mathbf{m}}$, making them suitable for *multi-bit* watermark schemes.

For *zero-bit* watermark systems, such as Tree-Ring Wen et al. (2023), the goal is not to transmit specific information but to detect the presence of a watermark through statistical test. Since no decodable bit sequence is embedded, traditional bit accuracy metrics are not directly applicable. Instead, methods like Tree-Ring use the *p-value* as a detection criterion: a *p-value* below a preset threshold indicates the presence of a watermark. However, this approach struggles to enable fair comparisons across all watermark models. Additionally, metrics like AUROC, while independent of specific threshold choices, do not guarantee high true positive rates (TPR) at low false positive rates (FPR). Relying solely on bit accuracy or AUROC may not suffice for achieving the desired performance in watermark detection An et al. (2024); Lu et al. (2024).

**TPR@$x$%FPR** is a more practical metric, as it evaluates both *zero-bit* and *multi-bit* watermark systems by measuring detection performance at an extremely low false positive rate. This metric better reflects the real-world usability and robustness of watermarking systems.

Let $\mathbf{x}$ denote an input image, which may or may not contain a watermark. To evaluate both *zero-bit* and *multi-bit* watermarking systems under a unified framework, a scalar detection statistic $T(\mathbf{x})$ is used. This statistic maps the detector's confidence into a comparable score across different watermarking paradigms.

For *zero-bit* watermarks (e.g., Tree-Ring Wen et al. (2023)), no explicit message is embedded. Detection relies on rejecting the null hypothesis of "no watermark." A natural choice is

$$T(\mathbf{x}) = -\log p(\mathbf{x}),$$

where $p(\mathbf{x})$ is the *p-value* under $\mathcal{H}_0$. Smaller $p$-values indicate stronger evidence against the null hypothesis and correspond to larger $T(\mathbf{x})$.

For *multi-bit* watermarks, a message $\mathbf{m}$ is embedded and decoded as $\hat{\mathbf{m}}$. The detection statistic $T(\mathbf{x})$ can be defined as the similarity between the embedded and decoded messages. Typical choices include bit accuracy, negative Hamming distance and etc.

A detection threshold $\theta$ is applied to $T(\mathbf{x})$ to decide whether a watermark is present. The threshold is chosen such that the false positive rate (FPR) is controlled at a desired level (e.g., 0.1%). Under this threshold, the corresponding true positive rate (TPR) can be measured. This yields the key metric TPR@$x$%FPR, which evaluates detection performance at extremely low false alarm rates for both zero-bit and multi-bit systems. Formally, consider the binary hypothesis test:

$$\mathcal{H}_0 : x \sim P_0 \text{ (cover image without watermark)},$$
$$\mathcal{H}_1 : x \sim P_1 \text{ (attacked protected image with watermark)}.$$

The detector declares the presence of a watermark if $T(\mathbf{x}) > \theta$. Then, the FPR and TPR are defined as

$$\mathrm{FPR}(\theta) = \mathbb{P}_{x \sim P_0}[T(x) > \theta], \mathrm{TPR}(\theta) = \mathbb{P}_{x \sim P_1}[T(x) > \theta].$$

In practice, with $N_0$ samples under $\mathcal{H}_0$ and $N_1$ samples under $\mathcal{H}_1$, the empirical estimates are

$$\widehat{\mathrm{FPR}}(\theta) = \frac{1}{N_0}\sum_{i=1}^{N_0}\mathbb{I}(T(x_i^{(0)}) > \theta), \widehat{\mathrm{TPR}}(\theta) = \frac{1}{N_1}\sum_{i=1}^{N_1}\mathbb{I}(T(x_i^{(1)}) > \theta).$$

The threshold $\theta^*$ for a target false positive rate $x$% is chosen as

$$\theta^* = \min\left\{\theta : \widehat{\mathrm{FPR}}(\theta) \leq \frac{x}{100}\right\},$$

and the corresponding true positive rate defines the metric

$$\mathrm{TPR@}x\mathrm{\%FPR} = \widehat{\mathrm{TPR}}(\theta^*).$$

This metric allows fair and direct comparison between fundamentally different watermarking paradigms. Zero-bit systems are evaluated by their ability to correctly reject $\mathcal{H}_0$ at a low false positive rate (FPR), while multi-bit systems are evaluated by the fraction of correctly decoded messages under the same constraint. A detection threshold $\theta^*$ is applied to the statistic $T(\mathbf{x})$ to decide whether a watermark is present. The threshold $\theta^*$ is chosen as the minimum value that ensures the FPR is fixed at the target level (e.g., 0.1%), and the corresponding true positive rate (TPR) is then measured. Unlike AUROC, which averages performance over all thresholds and may obscure behavior at very low FPRs, or raw bit accuracy, which ignores false alarms, TPR@$x$%FPR explicitly measures real-world operational performance under a controlled FPR.

**RQ-AUC:** To provide a more detailed ranking of all watermarking models, we further evaluate their robustness. Since TPR@$x$%FPR applies to both *multi-bit* and *zero-bit* watermarks, we define a new robustness score based on this metric. Specifically, under the same type of distortion, a watermarking model is usually subjected to attacks of varying strengths. Each attack strength

corresponds to a $\mathrm{TPR}@x\%\mathrm{FPR}$ value and also causes distortion to the protected image, measured by the PSNR between the protected image and the attacked image. Therefore, for each type of distortion, we can obtain a $\mathrm{TPR}@x\%\mathrm{FPR}$ vs. PSNR curve. For clarity, we denote this relationship as: $\mathrm{TPR}@x\%\mathrm{FPR} = f(\mathrm{PSNR})$, which represents the TPR at a fixed $x\%$ FPR as a function of the normalized PSNR. We then compute the area under this curve (AUC) to quantify the robustness of a watermarking model, therefore, a larger area indicates stronger robustness while preserving higher image quality. Since $\mathrm{TPR}@x\%\mathrm{FPR}$ primarily measures the *robustness* of the watermark and PSNR reflects the post-attack image *quality*, we name this metric *RQ-AUC* (Robustness-Quality AUC). The calculation is defined as follows:

$$\mathrm{RQ\text{-}AUC} = \int_0^{\mathrm{PSNR_{MAX}}} f(\mathrm{PSNR}) \, d(\mathrm{PSNR}). \tag{4}$$

where $\mathrm{PSNR_{MAX}}$ is the maximum PSNR across all the attacked protected image.

### A.3 Attacker Efficiency Metrics

$\mathbf{PSNR}@(\mathbf{TPR}@x\%\mathbf{FPR} = r)$: From the attacker's perspective, the goal is to reduce watermark detectability while preserving the visual quality of the attacked image. Therefore, attack efficiency depends not only on the extent to which watermark robustness is broken but also on the distortion introduced by the attack. The robustness is typically measured using robustness metrics such as $\mathrm{TPR}@x\%\mathrm{FPR}$, while the visual quality can be qualified by PSNR. However, not all attackers can achieve arbitrary levels of $\mathrm{TPR}@x\%\mathrm{FPR}$. For example, for some attackers, even under the maximum attack strength, the achievable effectiveness may only reach $\mathrm{TPR}@x\%\mathrm{FPR} = 0.9$. In such cases, higher requirements for $\mathrm{TPR}@x\%\mathrm{FPR}$ cannot be satisfied. To address this, we apply $\mathrm{PSNR}@(\mathrm{TPR}@x\%\mathrm{FPR} = r)$ as an evaluation metric for attack efficiency. Specifically, under a given attack requirement $r$, attack efficiency $E$ is defined as:

$$E = \mathrm{PSNR} \mid (\mathrm{TPR}@x\%\mathrm{FPR} = r),$$

Figure 12: Example of attacker efficiency measured by $\mathrm{PSNR}@(\mathrm{TPR}@x\%\mathrm{FPR} = r)$.

where a higher PSNR at the target rate $r$ indicates greater attack efficiency. In other words, the attacker achieves the desired reduction in watermark detectability while maintaining better visual quality. As shown in Figure 12, for each attacker, this efficiency is represented by the intersection of its $\mathrm{PSNR}$–$\mathrm{TPR}@x\%\mathrm{FPR}$ curve and the horizontal line $\mathrm{TPR}@x\%\mathrm{FPR} = r$. The PSNR value at this intersection quantifies the minimal visual degradation needed to meet the attack goal. If no intersection exists, the attacker cannot reach the target level $r$, even at maximum distortion.

## B Transferability Against Regeneration by Diffusion Models

Although the VINE Lu et al. (2024) paper claims significant robustness against various image editing techniques, including regeneration, we find that the method lacks transferability. Specifically, we performed regeneration using `StableDiffusionV2.1` and `StableDiffusionV2.1-base`. The visual comparison results are shown in Figure 13. Based on $\mathrm{TPR}@0.1\%\mathrm{FPR}$, as shown in Figure 14, the experiments indicate that VINE is more robust to regeneration by `StableDiffusionV2.1` than by `StableDiffusionV2.1-base`. This demonstrates that VINE does not exhibit transferability against regeneration attacks.

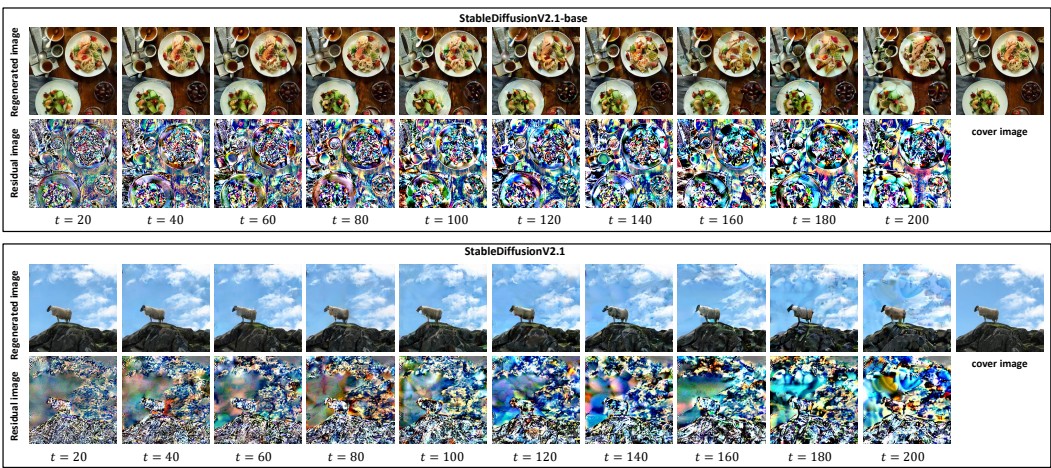

Figure 13: Visual comparison of images regenerated by `StableDiffusionV2.1` and `StableDiffusionV2.1-base` to evaluate the robustness of VINE. Differences in visual quality highlight the impact of regeneration with different diffusion models.

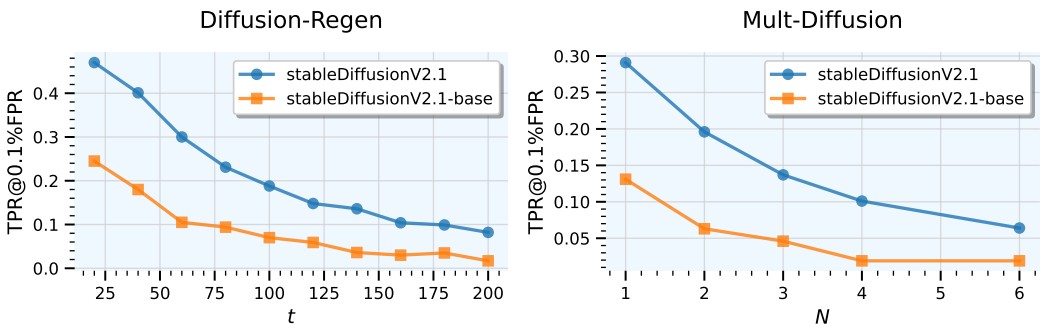

Figure 14: TPR@0.1%FPR of VINE Lu et al. (2024) under regeneration attacks using `StableDiffusionV2.1` and `StableDiffusionV2.1-base`. The results show higher robustness for `StableDiffusionV2.1`, indicating that VINE lacks transferability across different diffusion-based regeneration attacks.

## C  ADDITIONAL EXPERIMENTAL RESULTS

Figures 15 and 16 report the extraction accuracy and TPR@0.1%FPR of all *PGW* and *IGW* methods under 34 attacks, respectively. Figures 17 and 18 present the overall rankings of watermarking models and attackers based on the **RQ-AUC** metric. Figures 19 and 20 illustrate the PSNR vs. TPR@0.1%FPR and PSNR vs. extraction accuracy curves for all watermarking models, respectively. Finally, Figures 21–29 show the PSNR@(TPR@0.1%FPR = $r$)) results, where $r \in [0.1, 0.2, 0.3, 0.4, 0.5, 0.6, 0.7, 0.8, 0.9]$, for all models. Table 1 show all the 34 attacks. Together, these figures provide supplementary benchmark results that complement the analyses and discussions in the main paper.

## D  A SIMPLE GUIDE TO USING WATERMARKLAB

We have released WATERMARKLAB as a PyPI library, which currently integrates 3 *IGWs* (Tree-Ring Wen et al. (2023), GaussianShading Yang et al. (2024), StableSignature Fernandez et al. (2023)) and 6 *PGWs* (dctDwtSvd Cox et al. (2007), RivaGAN Zhang et al. (2019), StegaStamp Tancik et al. (2020), TrustMark Bui et al. (2023a), InvisMark Xu et al. (2025), VINE Lu et al. (2024)) to support comparative studies and benchmarking. In the future, we plan to further extend WATERMARKLAB with additional watermarking methods. WATERMARKLAB provides a comprehensive API that enables researchers to implement custom datasets, watermarking methods, attackers, evaluation metrics, and visualizations.

Figure 15: Extraction accuracy of all *PGW* and *IGW* methods under various attacks.

Figure 15: Extraction accuracy of all *PGW* and *IGW* methods under various attacks.

Figure 16: TPR@0.1%FPR of all *PGW* and *IGW* methods under various attacks.

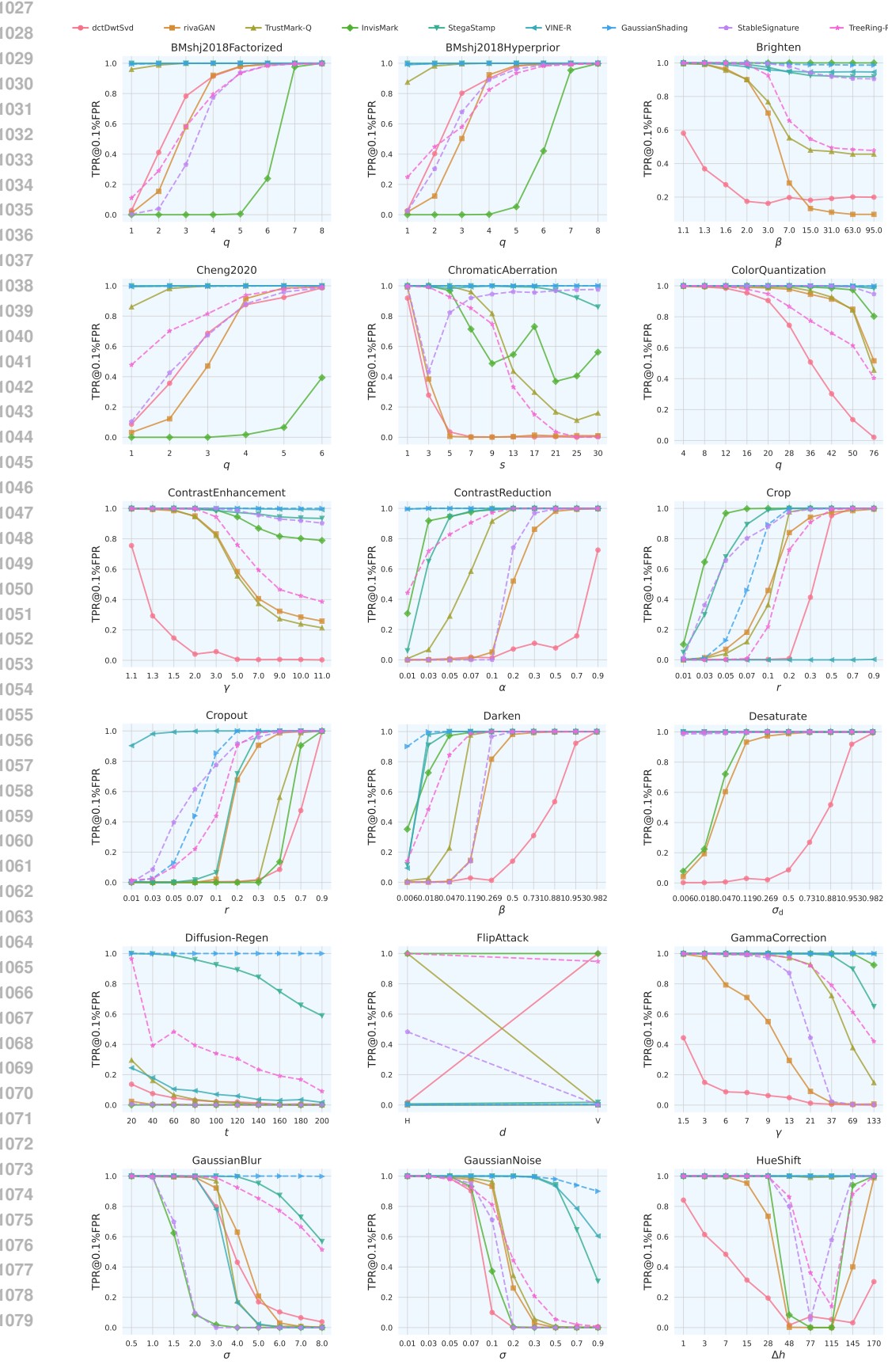

Figure 16: TPR@0.1%FPR under individual attackers (Continued).

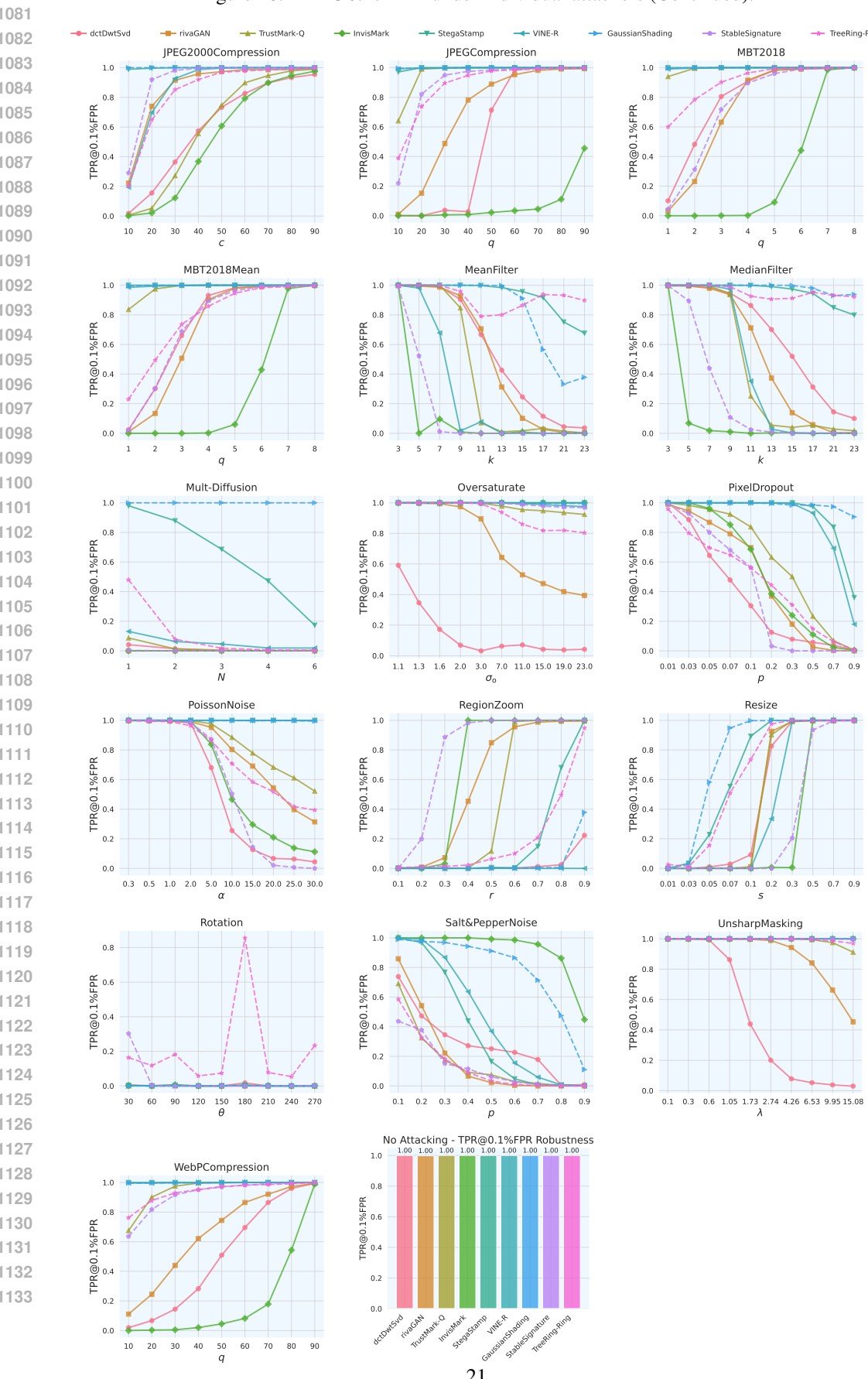

Figure 17: Robustness ranking of all watermarking methods under each of the 34 individual attackers by using **RQ-AUC**.

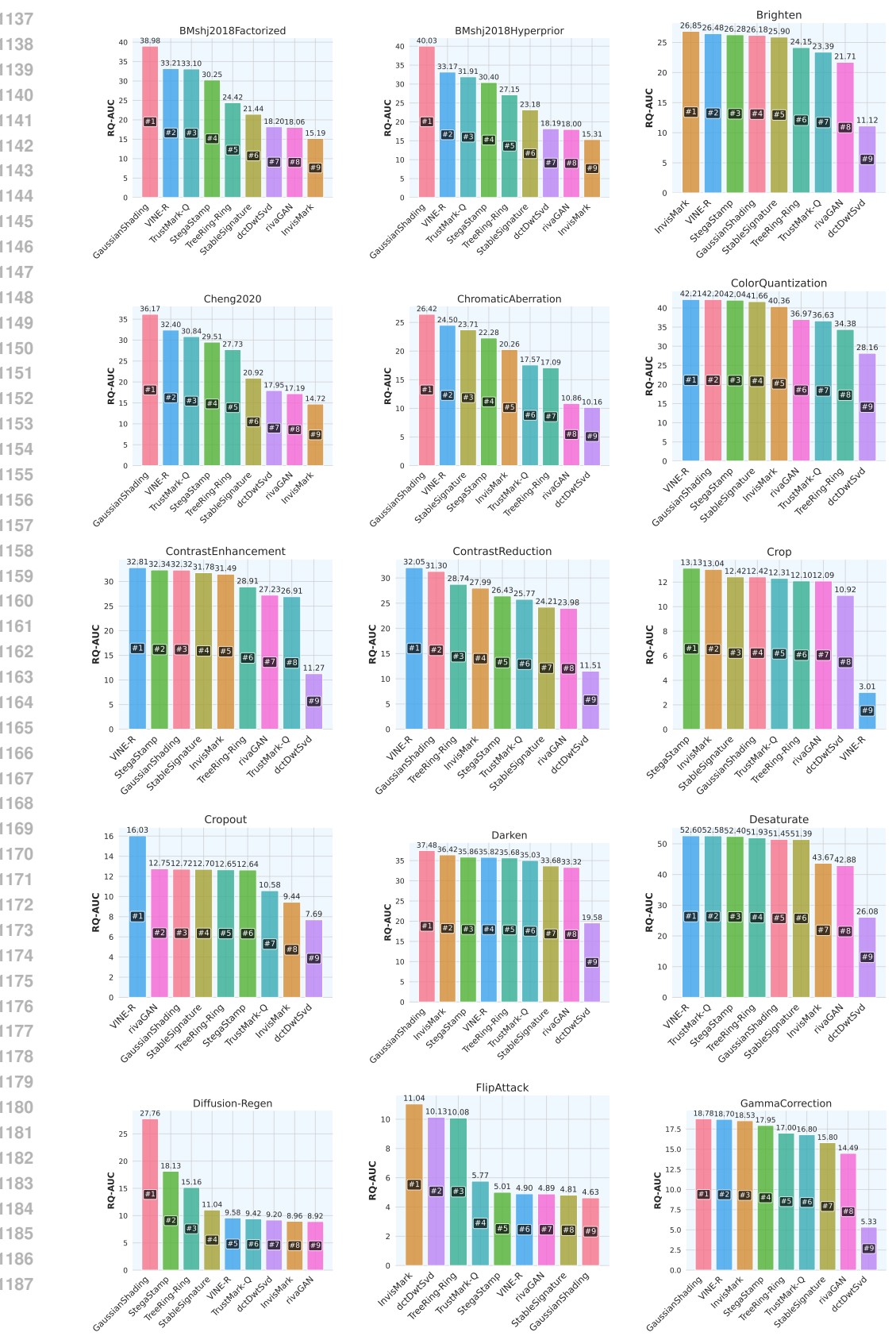

Figure 17: Robustness ranking of all watermarking methods under each of the 34 individual attackers by using **RQ-AUC**.

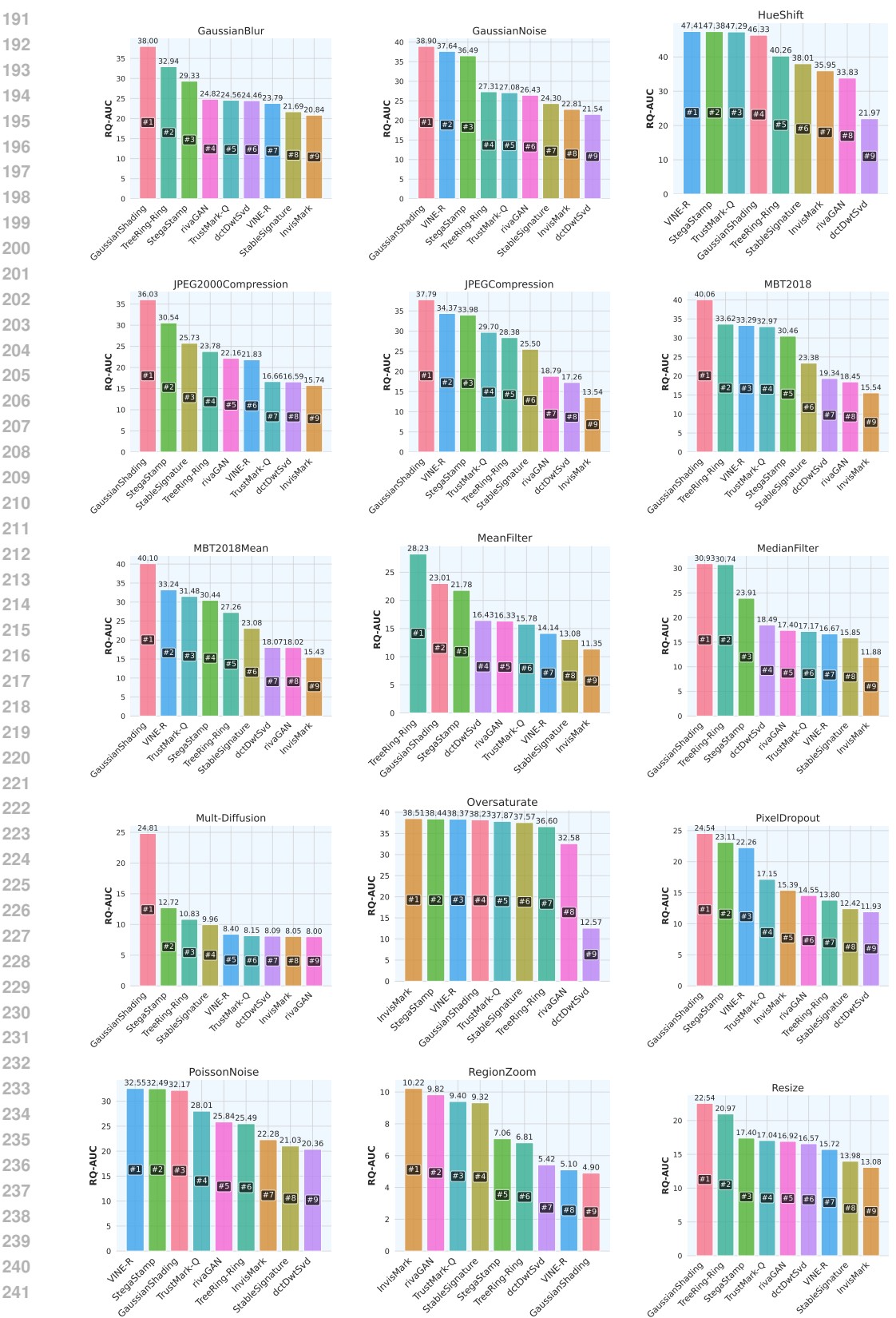

Figure 17: Robustness ranking of all watermarking methods under each of the 34 individual attackers by using **RQ-AUC**.

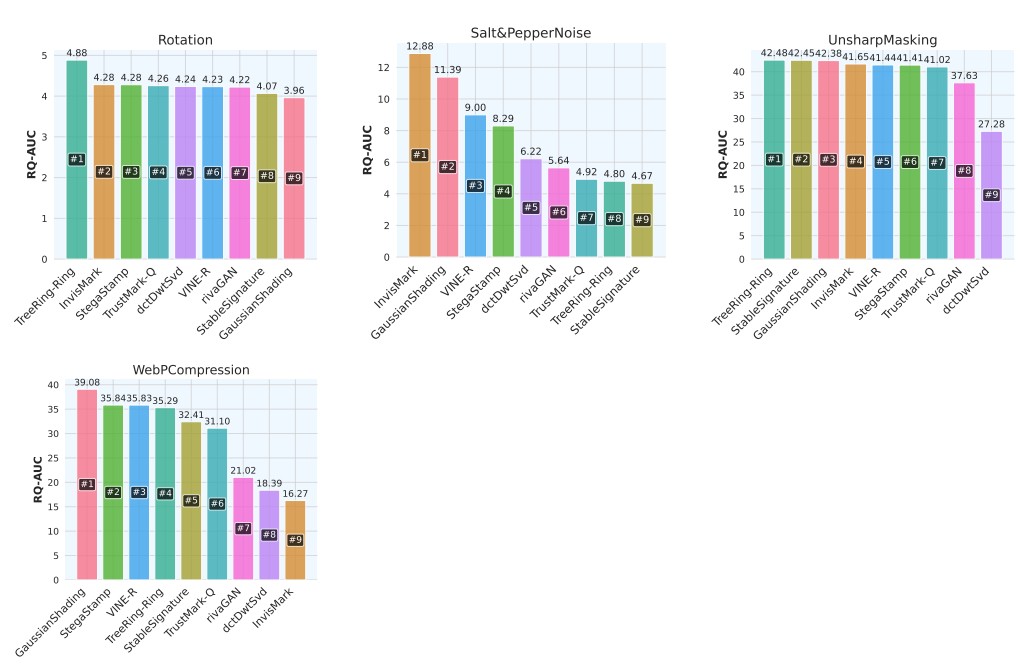

Figure 18: Robustness ranking of all watermarking methods under each of the 34 individual attackers by using **RQ-AUC**.

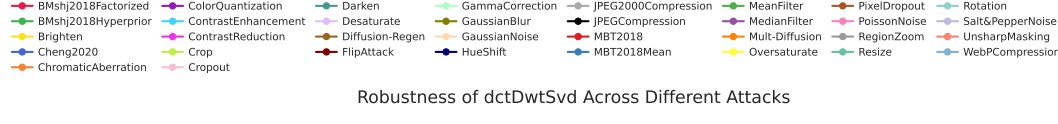

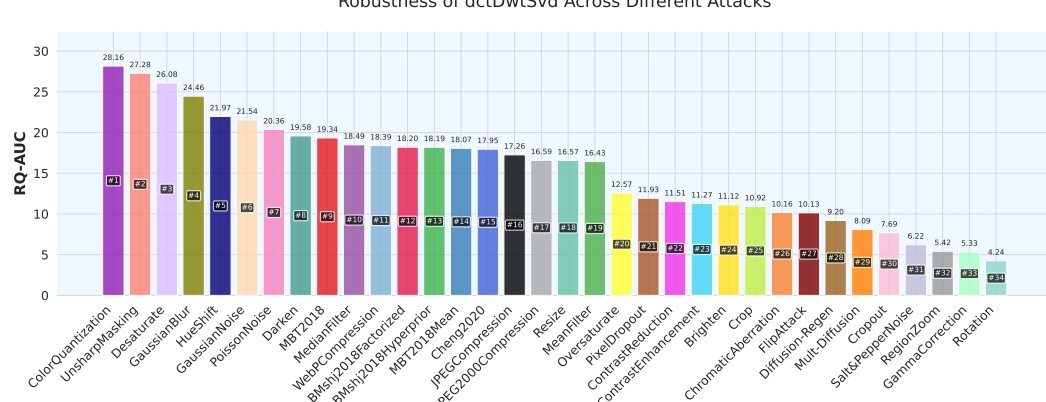

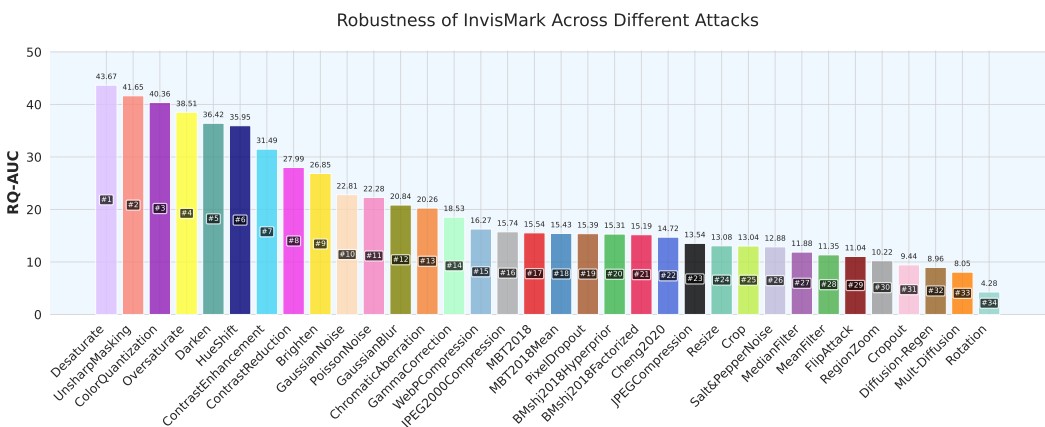

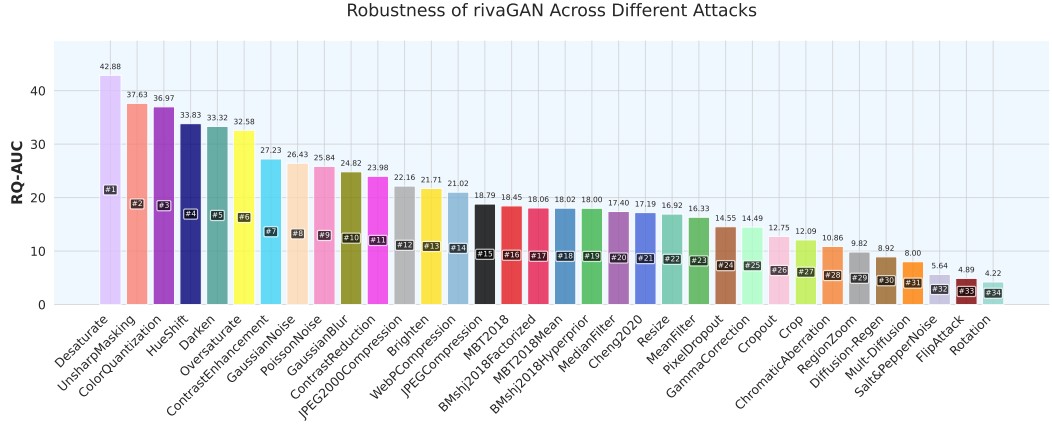

Figure 18: Robustness ranking of all watermarking methods under each of the 34 individual attackers by using **RQ-AUC**.

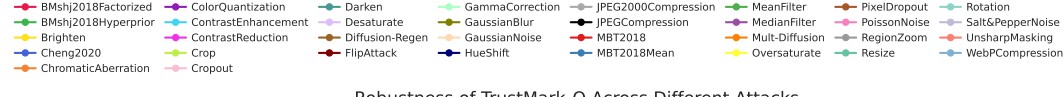

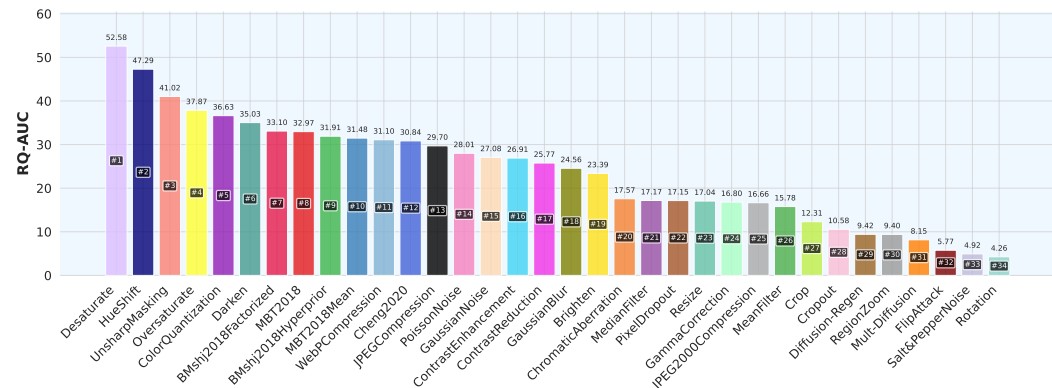

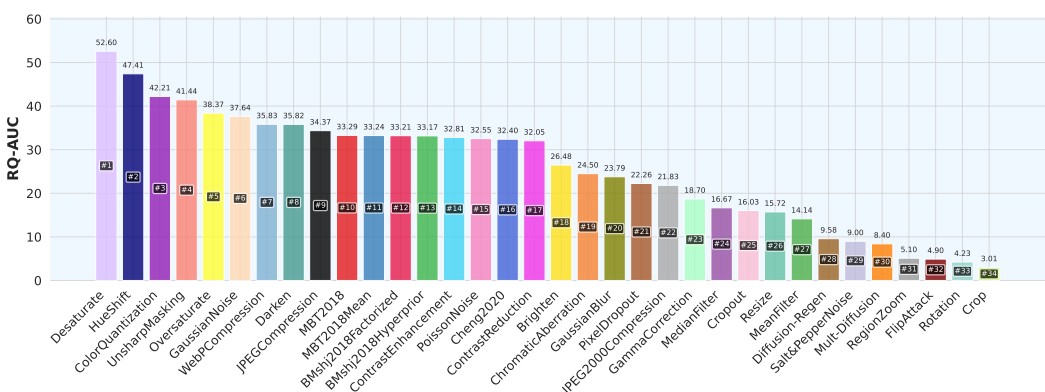

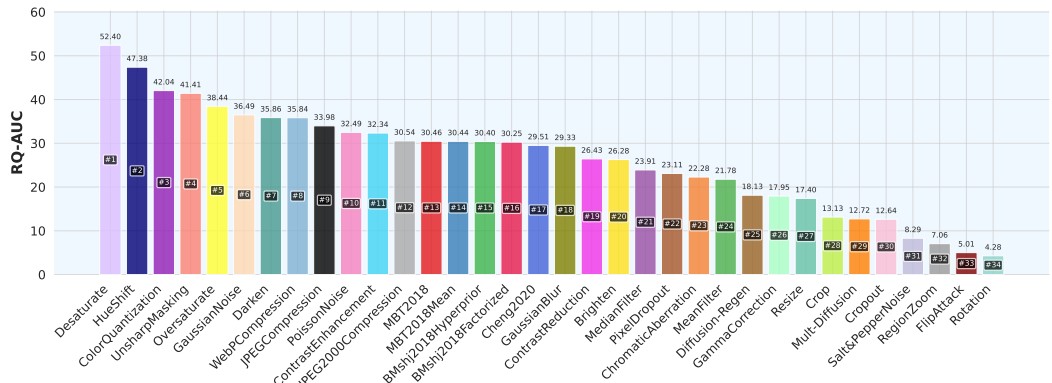

Figure 18: Robustness ranking of all watermarking methods under each of the 34 individual attackers by using **RQ-AUC**.

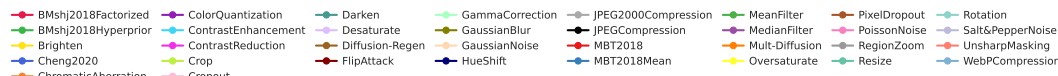

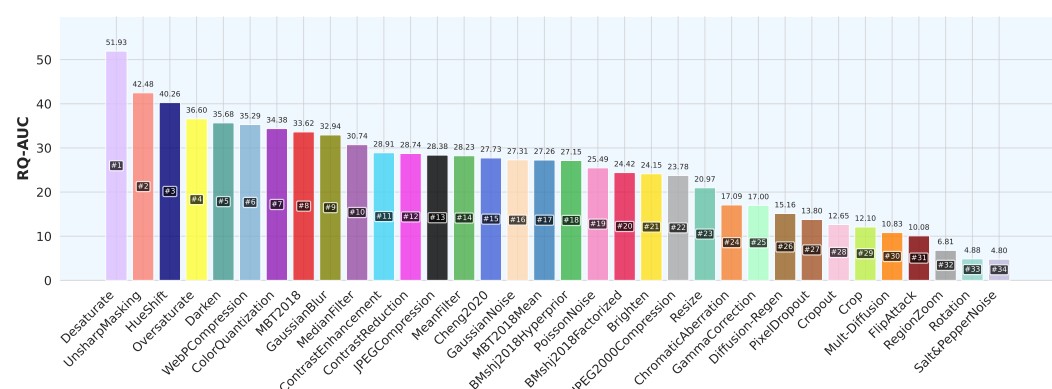

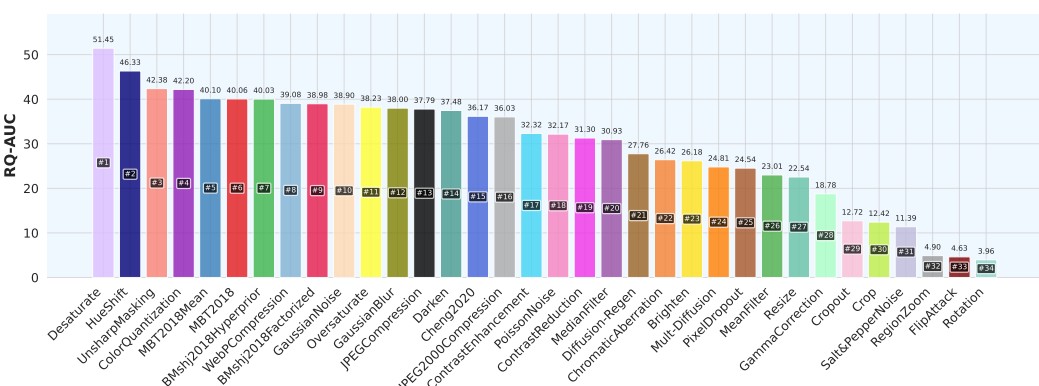

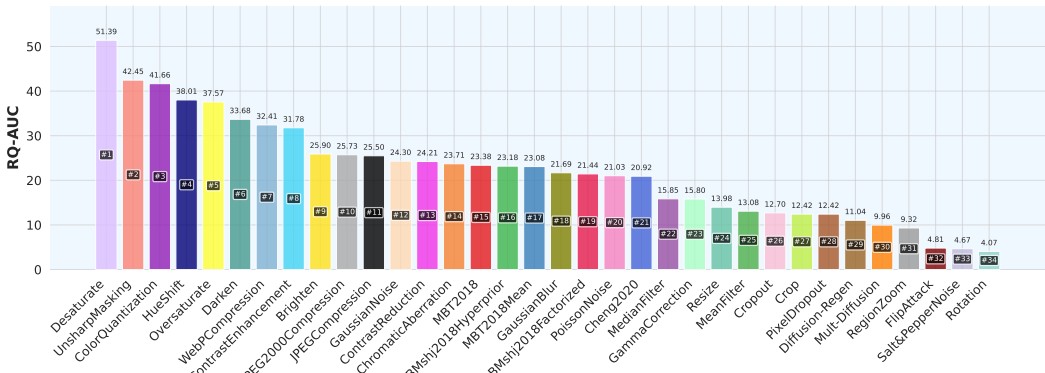

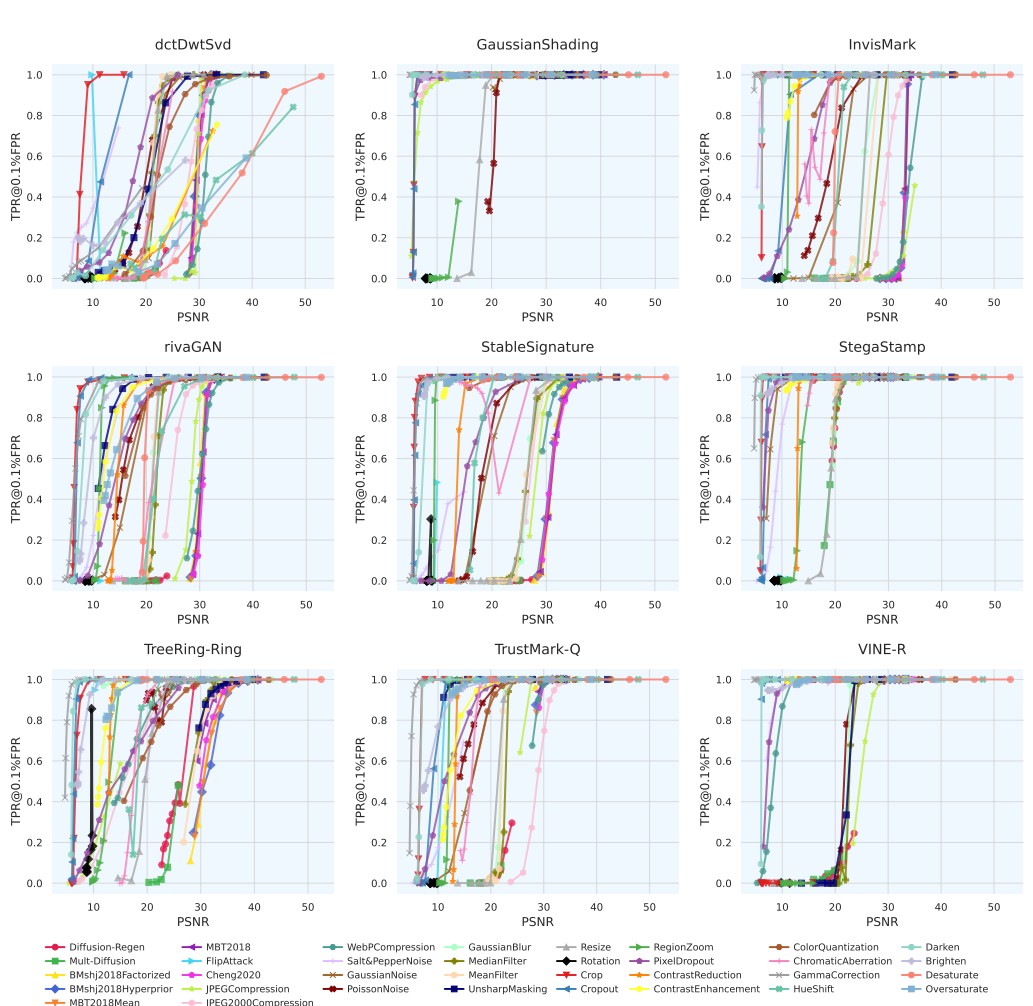

Figure 19: PSNR vs. TPR@$x$%FPR curves of all watermark models under 34 attacks. Curves closer to the bottom-right corner indicate stronger attacks, causing lower detection rates with less visual distortion.

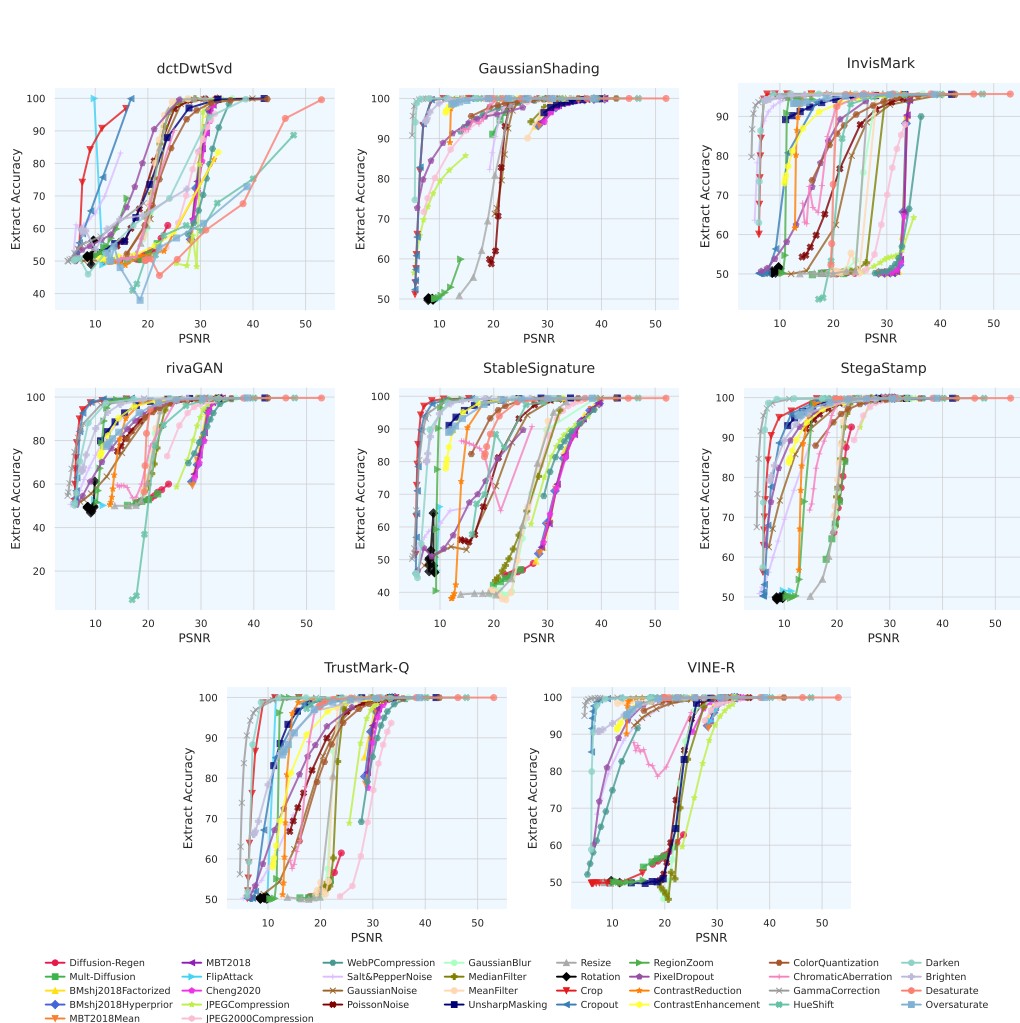

Figure 20: PSNR vs. Extract Accuracy curves of all watermark models under 34 attacks. Curves closer to the bottom-right corner indicate stronger attacks, causing lower detection rates with less visual distortion.

Figure 21: Attacker rankings for dctDwtSvd, measured by PSNR@(TPR@0.1%FPR=$r$). Higher PSNR indicates greater attack efficiency.

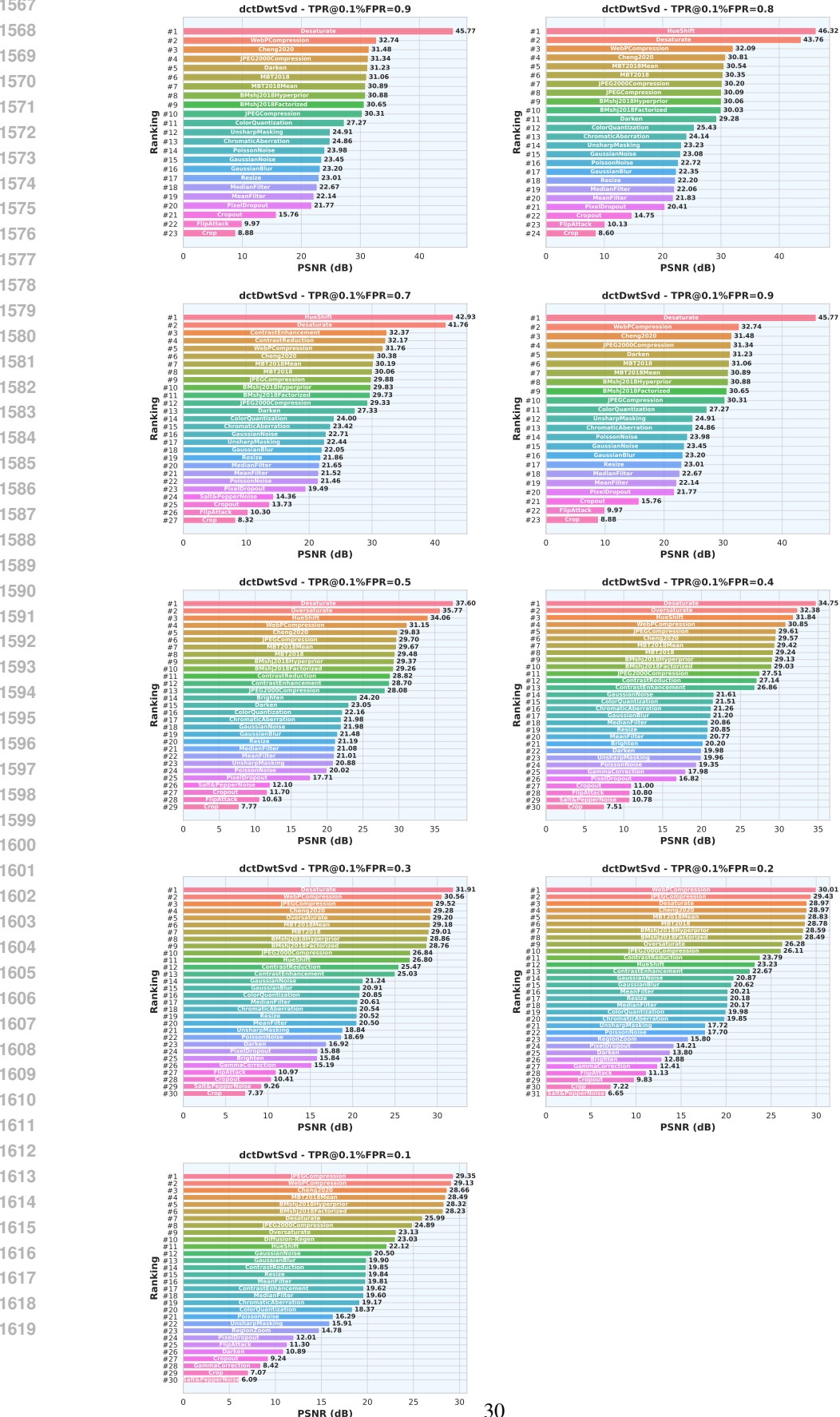

Figure 22: Attacker rankings for GaussianShading, measured by PSNR@(TPR@$0.1\%$FPR=$r$). Higher PSNR indicates greater attack efficiency.

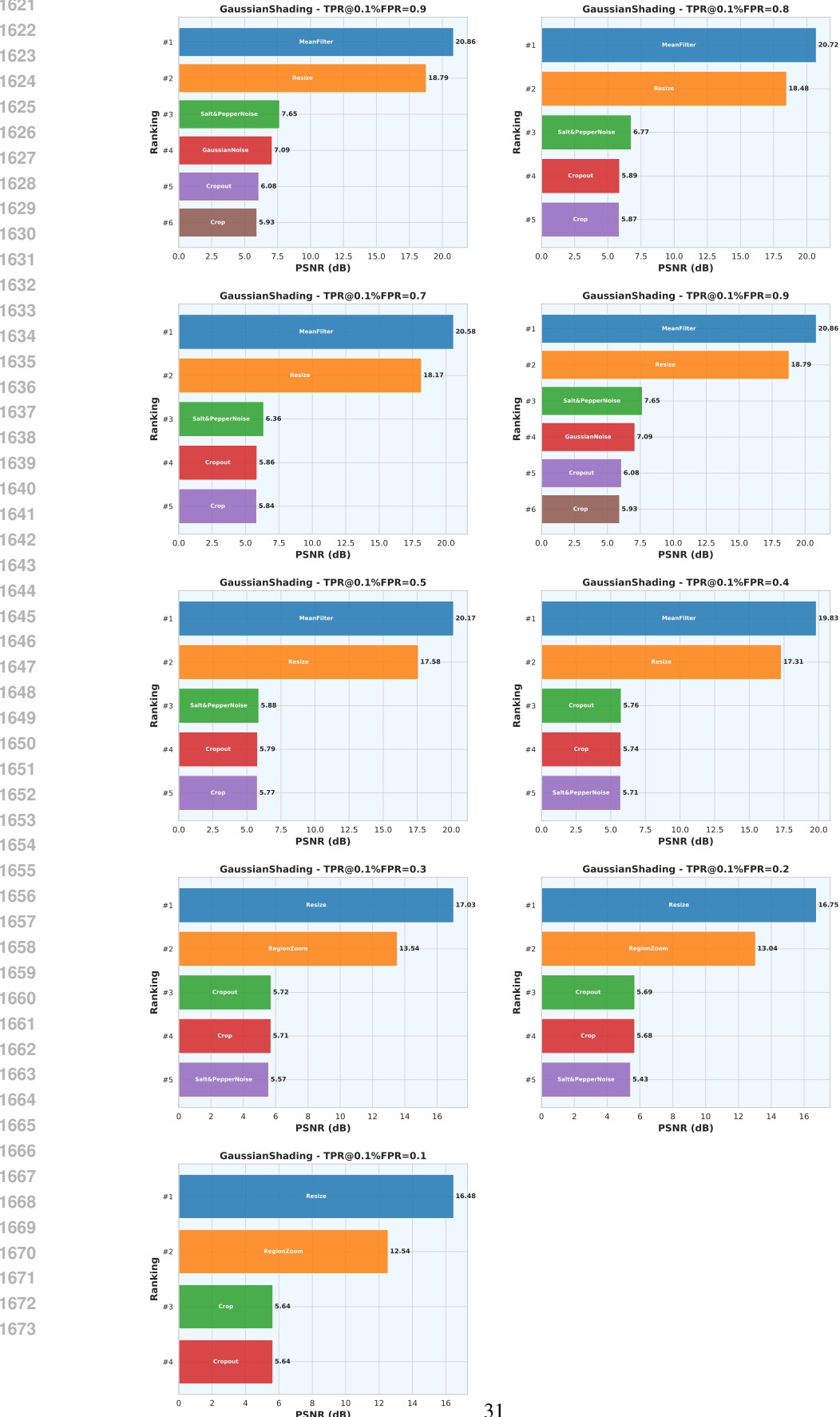

Figure 23: Attacker rankings for InvisMark, measured by PSNR@(TPR@0.1%FPR=$r$). Higher PSNR indicates greater attack efficiency.

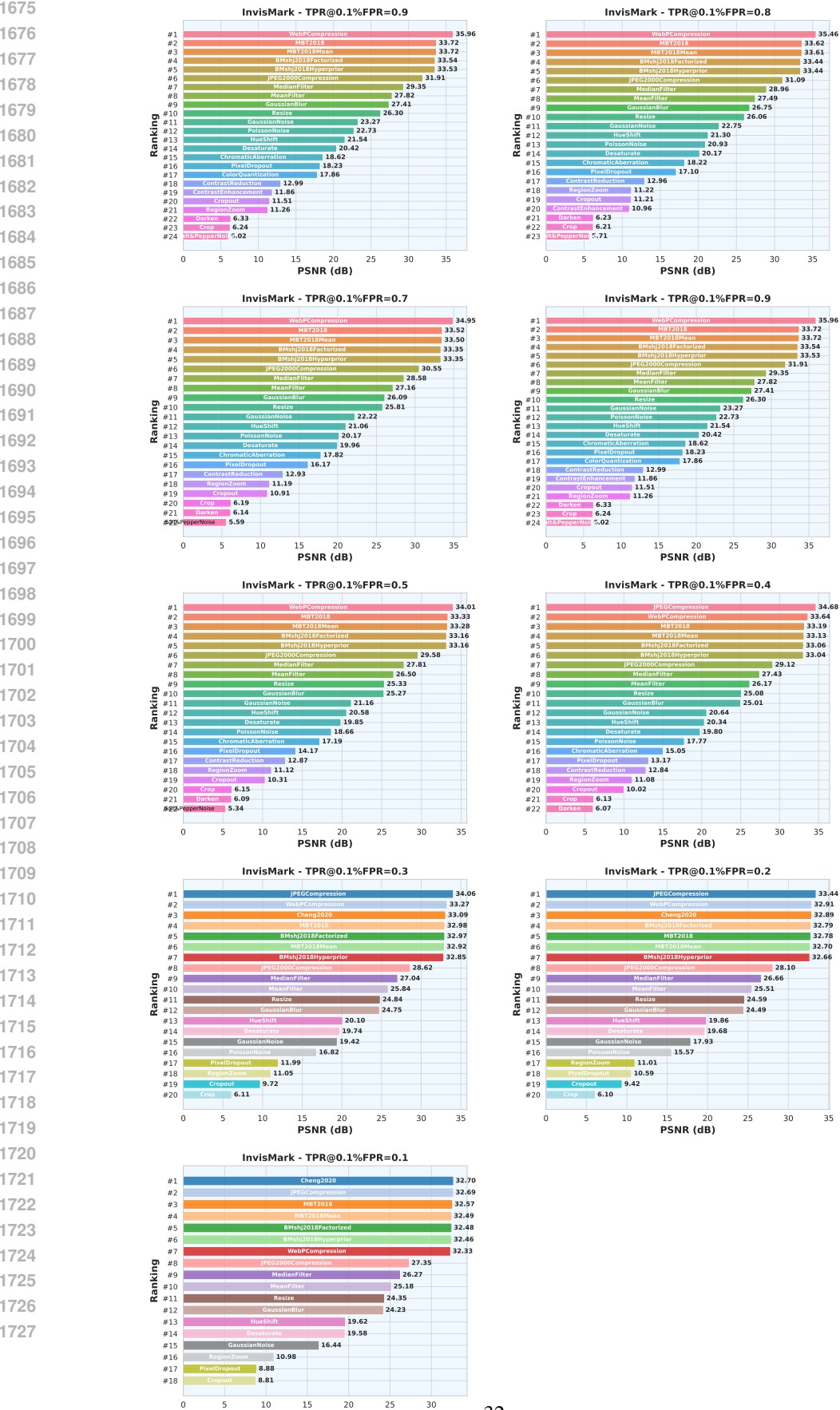

Figure 24: Attacker rankings for rivaGAN, measured by PSNR@(TPR@0.1%FPR=$r$). Higher PSNR indicates greater attack efficiency.

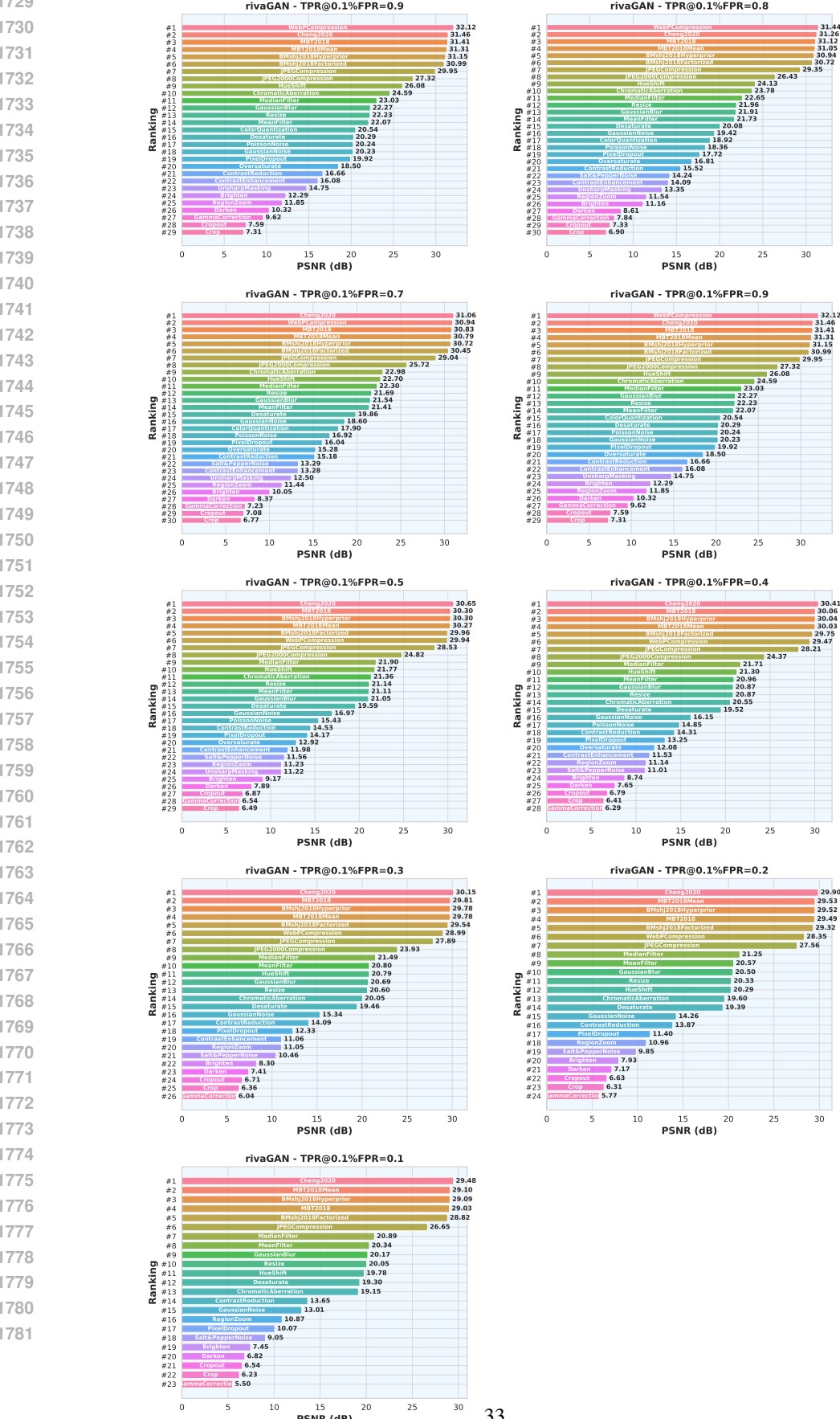

Figure 25: Attacker rankings for StableSignature, measured by PSNR@(TPR@0.1%FPR=$r$). Higher PSNR indicates greater attack efficiency.

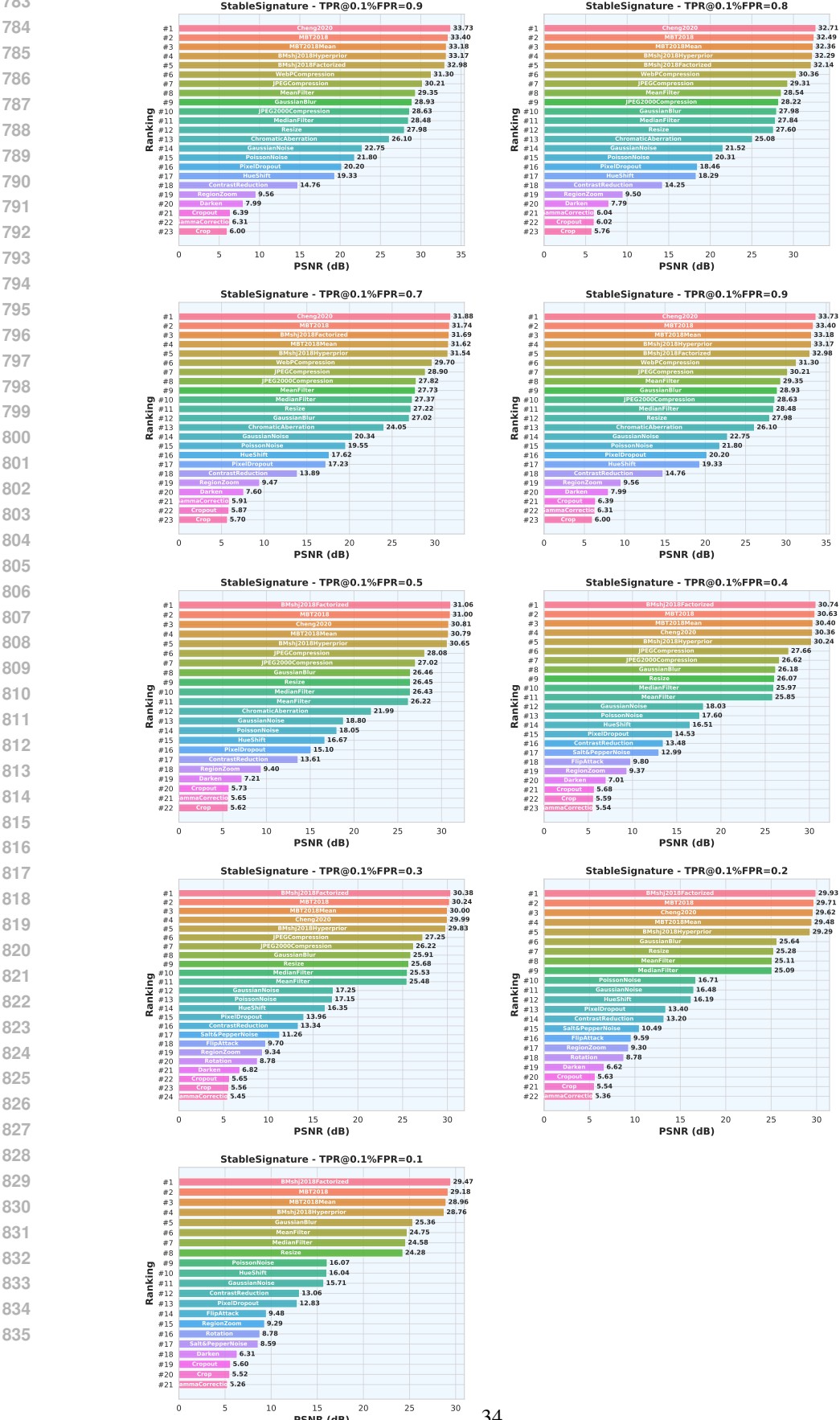

Figure 26: Attacker rankings for StegaStamp, measured by PSNR@(TPR@0.1%FPR=$r$). Higher PSNR indicates greater attack efficiency.

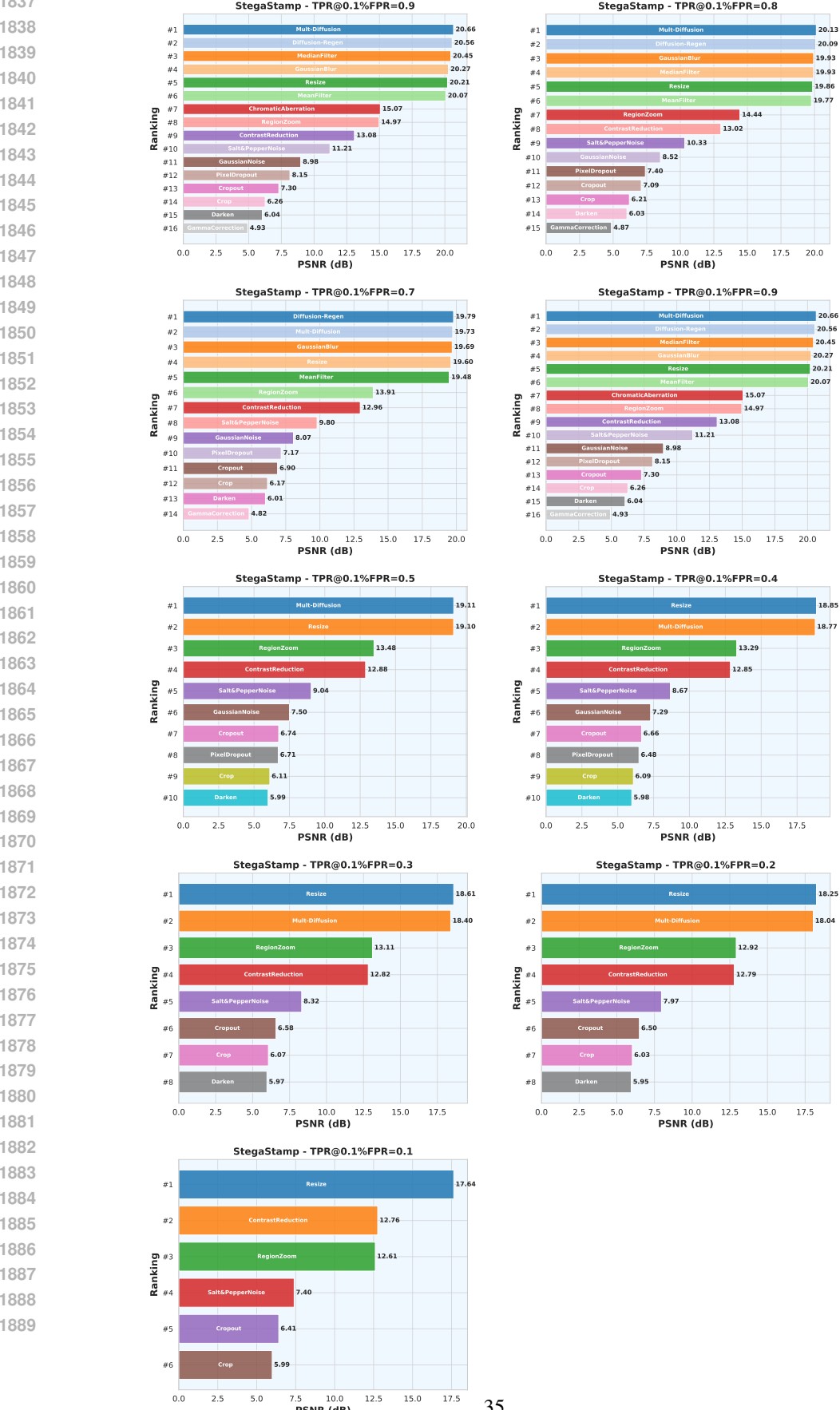

Figure 27: Attacker rankings for TreeRing-Ring, measured by PSNR@(TPR@0.1%FPR=$r$). Higher PSNR indicates greater attack efficiency.

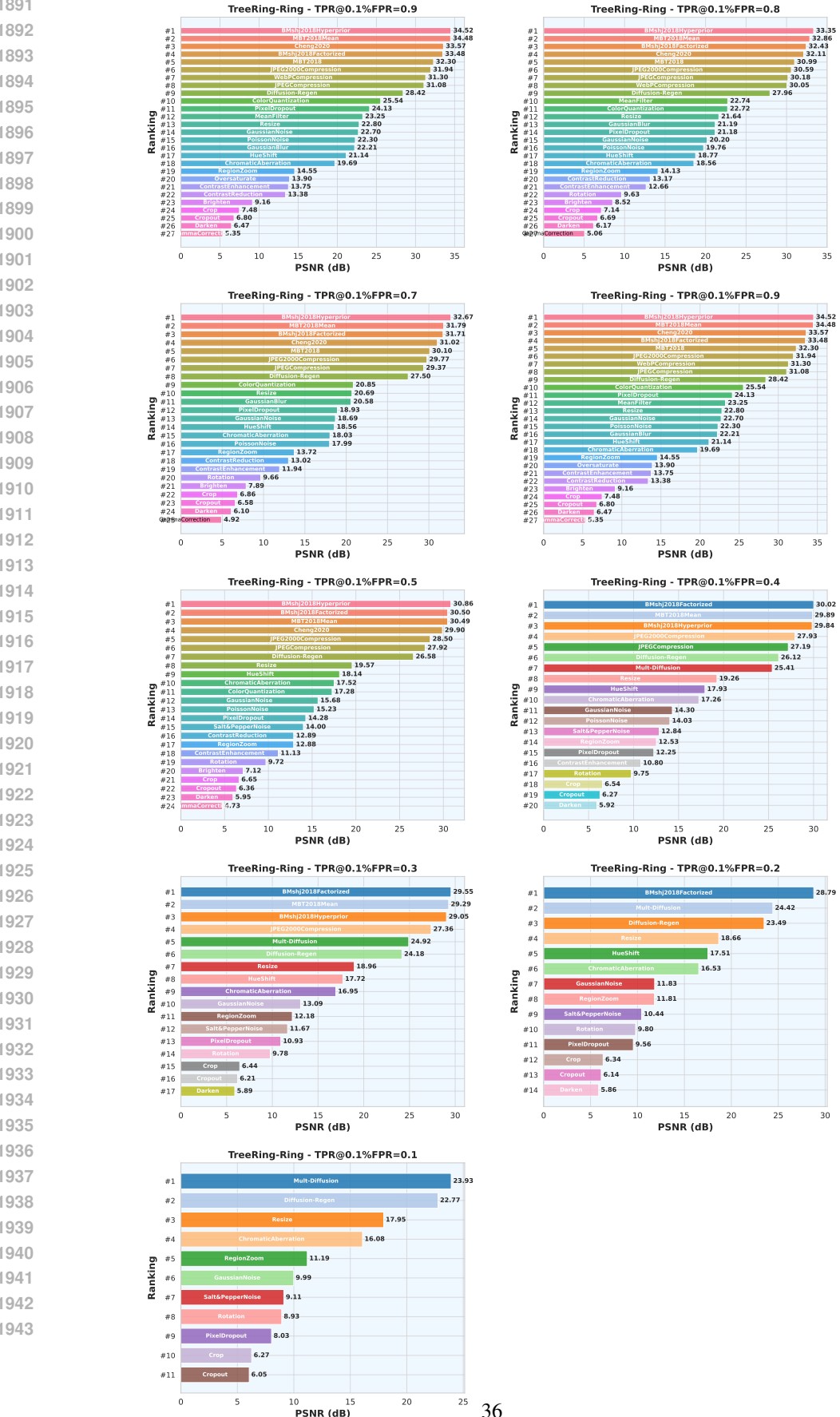

Figure 28: Attacker rankings for TrustMark-Q, measured by PSNR@(TPR@0.1%FPR=$r$). Higher PSNR indicates greater attack efficiency.

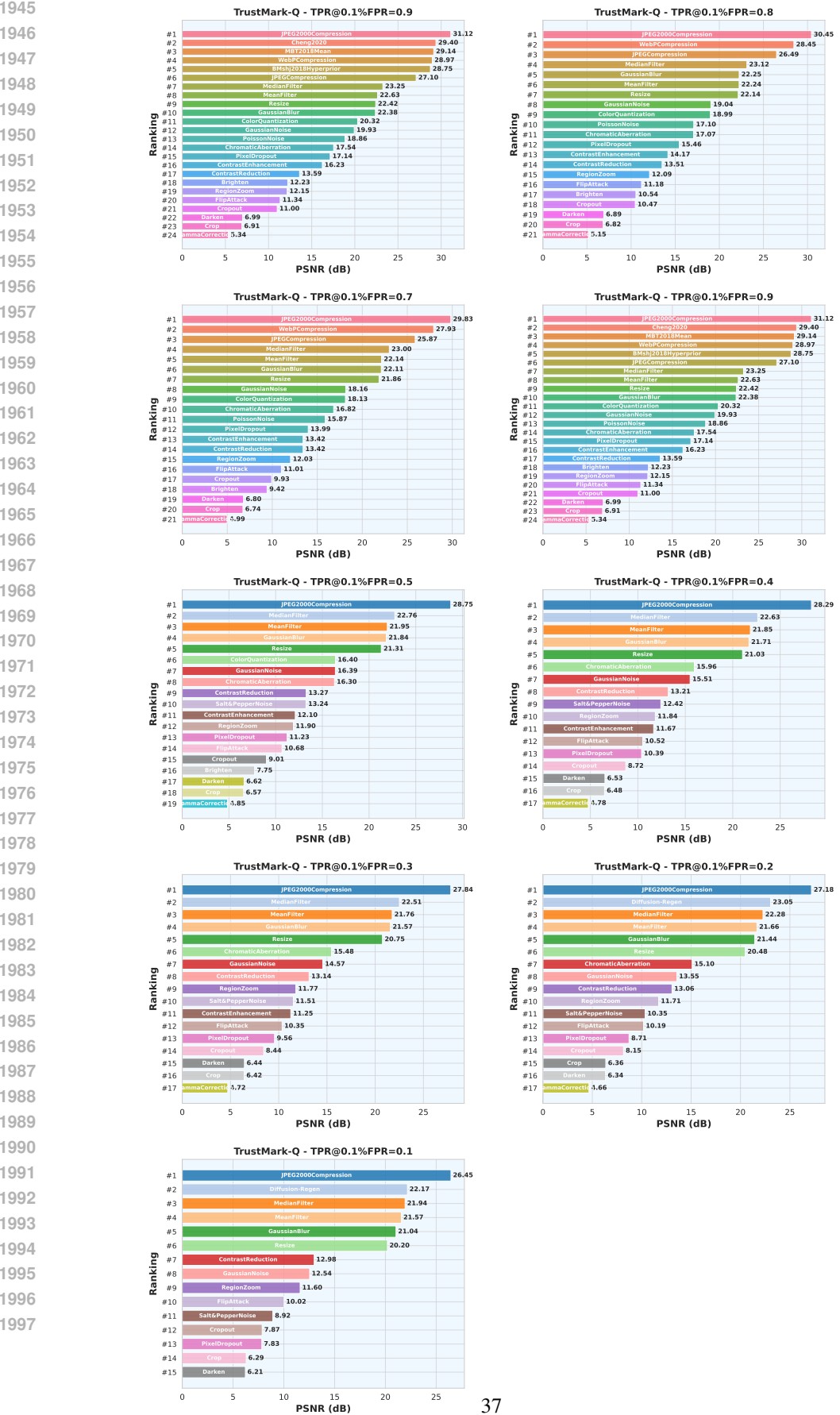

Figure 29: Attacker rankings for VINE-R, measured by PSNR@(TPR@$0.1\%$FPR=$r$). Higher PSNR indicates greater attack efficiency.

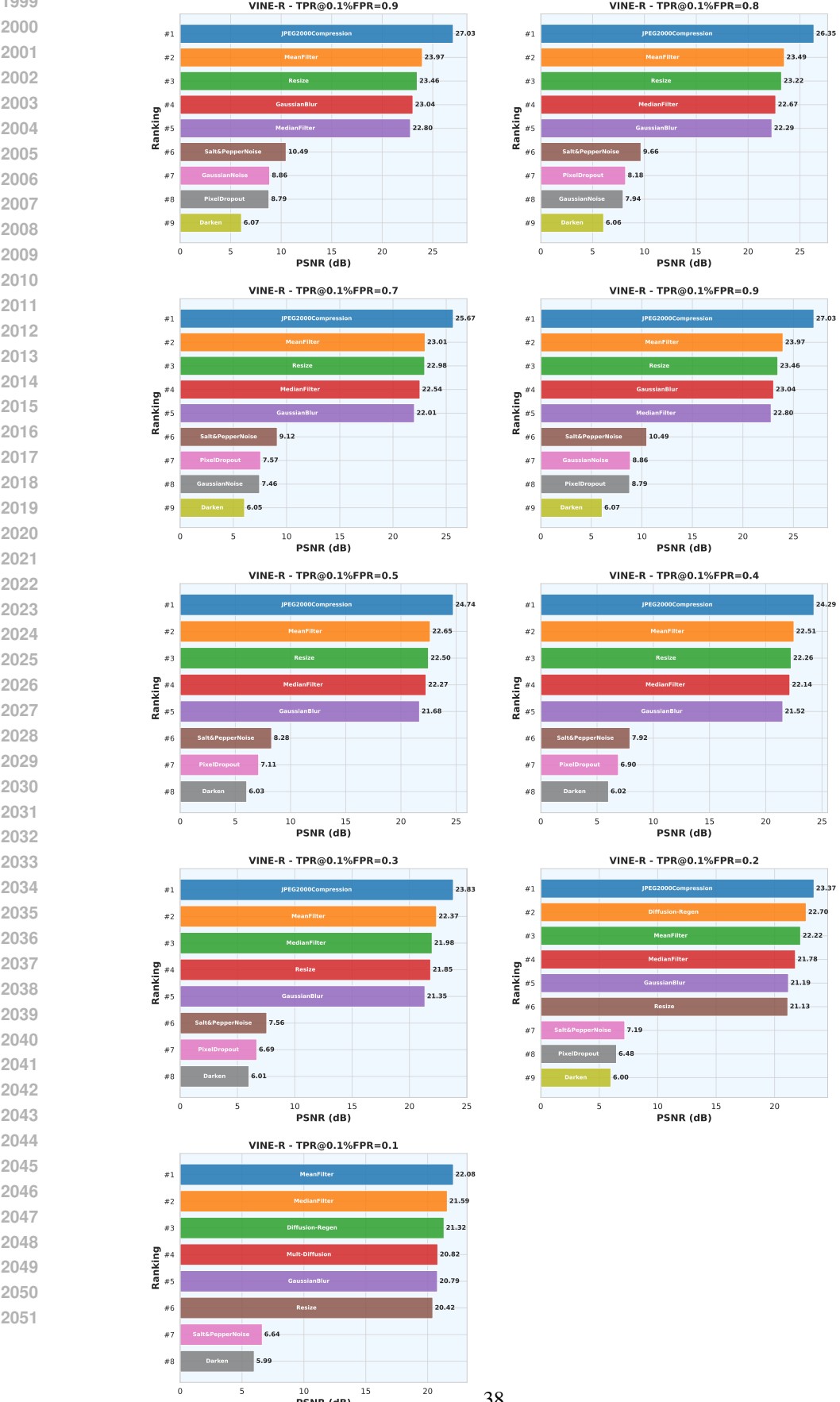

| Attack Name | Parameter Range |
|---|---|
| Diffusion-Regen | $t \in \{20, 40, 60, 80, 100, 120, 140, 160, 180, 200\}$ |
| Mult-Diffusion | $N \in \{1, 2, 3, 4, 6\}$, t=60 |
| BMshj2018Factorized | $q \in \{1, 2, 3, 4, 5, 6, 7, 8\}$ |
| BMshj2018Hyperprior | $q \in \{1, 2, 3, 4, 5, 6, 7, 8\}$ |
| MBT2018Mean | $q \in \{1, 2, 3, 4, 5, 6, 7, 8\}$ |
| MBT2018 | $q \in \{1, 2, 3, 4, 5, 6, 7, 8\}$ |
| Cheng2020 | $q \in \{1, 2, 3, 4, 5, 6\}$ |
| JPEGCompression | $q \in \{90, 80, 70, 60, 50, 40, 30, 20, 10\}$ |
| JPEG2000Compression | $c \in \{90, 80, 70, 60, 50, 40, 30, 20, 10\}$ |
| WebPCompression | $q \in \{90, 80, 70, 60, 50, 40, 30, 20, 10\}$ |
| Salt&PepperNoise | $p \in \{0.1, 0.2, 0.3, 0.4, 0.5, 0.6, 0.7, 0.8, 0.9\}$ |
| GaussianNoise | $\sigma \in \{0.01, 0.03, 0.05, 0.07, 0.1, 0.2, 0.3, 0.5, 0.7, 0.9\}$ |
| PoissonNoise | $\alpha \in \{30.0, 25.0, 20.0, 15.0, 10.0, 5.0, 2.0, 1.0, 0.5, 0.3\}$ |
| GaussianBlur | $\sigma \in \{0.5, 1.0, 1.5, 2.0, 3.0, 4.0, 5.0, 6.0, 7.0, 8.0\}$ |
| MedianFilter | $k \in \{3, 5, 7, 9, 11, 13, 15, 17, 21, 23\}$ |
| MeanFilter | $k \in \{3, 5, 7, 9, 11, 13, 15, 17, 21, 23\}$ |
| UnsharpMasking | $\lambda \in \{0.1, 0.3, 0.6, 1.05, 1.73, 2.74, 4.26, 6.53, 9.95, 15.08\}$ |
| Resize | $s \in \{0.01, 0.03, 0.05, 0.07, 0.1, 0.2, 0.3, 0.5, 0.7, 0.9\}$ |
| Rotation | $\theta \in \{30, 60, 90, 120, 150, 180, 210, 240, 270\}$ |
| FlipAttack | $d \in \{\mathrm{H}, \mathrm{V}\}$ |
| Crop | $r \in \{0.01, 0.03, 0.05, 0.07, 0.1, 0.2, 0.3, 0.5, 0.7, 0.9\}$ |
| Cropout | $r \in \{0.01, 0.03, 0.05, 0.07, 0.1, 0.2, 0.3, 0.5, 0.7, 0.9\}$ |
| RegionZoom | $r \in \{0.1, 0.2, 0.3, 0.4, 0.5, 0.6, 0.7, 0.8, 0.9\}$ |
| PixelDropout | $p \in \{0.01, 0.03, 0.05, 0.07, 0.1, 0.2, 0.3, 0.5, 0.7, 0.9\}$ |
| ContrastReduction | $\alpha \in \{0.01, 0.03, 0.05, 0.07, 0.1, 0.2, 0.3, 0.5, 0.7, 0.9\}$ |
| ContrastEnhancement | $\gamma \in \{1.1, 1.3, 1.5, 2.0, 3.0, 5.0, 7.0, 9.0, 10.0, 11.0\}$ |
| ColorQuantization | $q \in \{4, 8, 12, 16, 20, 28, 36, 42, 50, 76\}$ |
| ChromaticAberration | $s \in \{1, 3, 5, 7, 9, 13, 17, 21, 25, 30\}$ |
| GammaCorrection | $\gamma \in \{1.5, 3, 6, 7, 9, 13, 21, 37, 69, 133\}$ |
| HueShift | $\Delta h \in \{1, 3, 7, 15, 28, 48, 77, 115, 145, 170\}$ |
| Darken | $\beta \in \{0.006, 0.018, 0.047, 0.119, 0.269, 0.5, 0.731, 0.881, 0.953, 0.982\}$ |
| Brighten | $\beta \in \{1.1, 1.3, 1.6, 2.0, 3.0, 7.0, 15.0, 31.0, 63.0, 95.0\}$ |
| Desaturate | $\sigma_{\mathrm{d}} \in \{0.006, 0.018, 0.047, 0.119, 0.269, 0.5, 0.731, 0.881, 0.953, 0.982\}$ |
| Oversaturate | $\sigma_{\mathrm{o}} \in \{1.1, 1.3, 1.6, 2.0, 3.0, 7.0, 11.0, 15.0, 19.0, 23.0\}$ |

Table 1: Attacker list used for benchmarking, detailing the name of each attack method and the attack strengths.

