# OpenReview forum: "WatermarkLab: A Comprehensive Framework for Robust Image Watermarks Benchmarking and Development"
_ICLR.cc/2026/Conference — Submitted to ICLR 2026_

### Official Review · Reviewer_cXoi · 2025-10-17

**Soundness:** 2
**Presentation:** 3
**Contribution:** 2
**Rating:** 4
**Confidence:** 3

**Summary:**

This paper presents WatermarkLab, a comprehensive framework for benchmarking and developing robust image watermarking methods. The framework supports both in-generation watermarking (IGW) and post-generation watermarking (PGW), integrating 10 representative watermarking methods, 34 test attackers for evaluation, and 28 differentiable attackers for adversarial training. The authors conduct extensive experiments evaluating 9 watermarking schemes under 34 attacks, finding that GaussianShading and StegaStamp remain the most robust methods for IGW and PGW respectively, though both exhibit vulnerabilities to geometric distortions and regeneration attacks. The framework provides visualization tools and an interactive website for result analysis.

**Strengths:**

- The framework integrates diverse watermarking paradigms (zero-bit, multi-bit, IGW, PGW, robust reversible) with extensive attack coverage spanning compression, color transformations, geometric distortions, noise, diffusion-based regeneration, and filtering.

- The introduction of TPR@x%FPR as a metric that applies to both zero-bit and multi-bit watermarks, while the RQ-AUC metric (area under TPR@x%FPR vs. PSNR curve) provides a principled way to quantify robustness-quality tradeoffs across different attack types.

- The inclusion of 28 differentiable attackers for adversarial training, auxiliary tools (arithmetic coding, reversible data hiding), and modular architecture with clear base classes facilitates development of new watermarking methods beyond pure benchmarking.

**Weaknesses:**

- The paper lacks theoretical justification for why IGW methods fundamentally outperform PGW methods (GaussianShading achieves 1013.98 cumulative RQ-AUC vs. StegaStamp's 899.99). While the authors attribute this to "stronger implicit correlation with image content," no formal analysis or information-theoretic treatment is provided to explain the performance gap.

- The framework provides extensive empirical comparisons but limited guidance on architectural choices, e.g., why certain encoder-decoder designs work better, how watermark capacity affects robustness-quality tradeoffs, or optimal strategies for combining differentiable attackers during training.
All experiments use MS-COCO 2017 with limited exploration of domain generalization. No evaluation on diverse content types (medical images, satellite imagery, artistic styles) or resolution variations that would stress-test real-world applicability of these methods.

- No reporting of training time, inference latency, memory requirements, or model sizes across the 10 integrated methods. This omission is critical for practitioners choosing methods for deployment, especially for real-time applications or resource-constrained environments.

- Despite claiming comprehensiveness, the framework omits several important attack categories: adversarial perturbations specifically targeting watermark removal, watermark overwriting attacks, deep learning-based inpainting attacks beyond diffusion models, and physical-world attacks beyond print-capture (e.g., projector-camera distortions, varying illumination).

- While the paper identifies vulnerabilities (e.g., geometric distortions for GaussianShading), it provides insufficient investigation into why these failures occur or potential mitigation strategies. The surprising finding that VINE fails on simple crops but resists deep editing deserves deeper mechanistic analysis.

**Questions:**

- Can the authors provide information-theoretic analysis or capacity-distortion bounds explaining why IGW methods fundamentally outperform PGW methods? What is the theoretical limit on robustness achievable by PGW methods that embed watermarks as post-hoc perturbations versus IGW methods that integrate watermarks during generation?

- How do these watermarking methods perform across different image domains (medical, satellite, artistic) and resolutions? Have you evaluated domain shift scenarios where models trained on natural images (MS-COCO) are applied to specialized domains, and what is the degradation in robustness?

- How robust are these methods against adaptive attacks where an adversary has white-box access to the watermark detector? Have you evaluated attacks that specifically optimize to maximize watermark removal while minimizing perceptual distortion, rather than using predefined transformations?

- What are the computational costs (training time, inference latency, memory) for each method? For practitioners deploying watermarking at scale (e.g., protecting millions of generated images), which methods offer the best robustness-efficiency tradeoff?

- Can deeper analysis be provided of why VINE exhibits such different behavior for crop versus cropout attacks (TPR drops to 0.01 with 10% boundary crop but remains robust to 90% cropout)? What does this reveal about watermark embedding strategies, and could this insight inform better method design?

- How about combining multiple watermarking methods (e.g., both IGW and PGW) to achieve complementary robustness properties? Could ensemble approaches mitigate individual method vulnerabilities while maintaining imperceptibility?

---

### Official Review · Reviewer_BR4i · 2025-10-24

**Soundness:** 2
**Presentation:** 2
**Contribution:** 1
**Rating:** 2
**Confidence:** 3

**Summary:**

In this paper, the authors introduce a new benchmark for evaluating image watermarks. To ensure the benchmark is comprehensive, fair, open, and extensible, it incorporates 10 representative watermarking methods, covering both in-generation and post-generation techniques. To assess and improve watermark robustness, the benchmark further includes 34 attack methods for evaluation and 28 differentiable attacks for training. Finally, it provides a comprehensive visualization toolkit to facilitate clearer and more informative analysis.

**Strengths:**

The paper is well-written, and I appreciate the authors’ effort in integrating a wide range of watermarking and attack methods into a single benchmark. The evaluation is notably more comprehensive than in prior work, as it encompasses a greater variety of both watermarking and attack techniques.

**Weaknesses:**

- My main concern lies in the contribution of this paper. The proposed benchmark appears to be highly similar to WAVES, and the work seems more like an extension of WAVES through the inclusion of additional watermarking and attack methods.
- The paper emphasizes that the proposed benchmark is comprehensive, fair, open, and extensible. While I agree that it is indeed more comprehensive, the meanings of “fair” and “open” are not clearly justified. For instance, aren’t benchmarks like WAVES also fair and open? Additionally, the claim of being extensible could be a valuable contribution, but the paper does not elaborate on the specific design choices that enable extensibility. It would be helpful for the authors to discuss these aspects in more detail and consider sharing pseudocode or the full benchmark code, as the current website link appears to include only the visualization component.
- The diffusion model used in the paper, Stable Diffusion v2.1-Base, is somewhat outdated compared to more recent models.

**Questions:**

- How flexible is the benchmark in incorporating new watermarking methods? Would it be straightforward to add them?

---

> ### Author Response · Authors · 2025-11-13
>
> Thank you very much for reviewing our manuscript. We appreciate your time and would like to clarify a few points regarding potential misunderstandings.
>
> **1. Relationship with WAVES**
>
> WAVES is undoubtedly a landmark contribution. It established the first fair and open benchmark for image watermarking, covering a wide range of attacks including common distortions, surrogate attacks, and adversarial attacks. This work has provided crucial inspiration for subsequent watermarking designs, especially in the context of generative models. We deeply respect WAVES as a model for our own efforts.
>
> However, WAVES has limitations for the long-term growth of the community. First, despite its rich attack suite, it lacks an extensible architecture and does not provide a unified, user-friendly API, which makes “one-stop” evaluation of new algorithms difficult. Second, it offers no tools for watermark *development*.
>
> In contrast, WATERMARKLAB is not just a benchmark. It is a **comprehensive framework** that supports both *development* and *evaluation*. For example, it provides 28 differentiable distortion layers for end-to-end training of PGW methods and includes utilities such as compression codecs and reversible data hiding interfaces to enable fair comparison in two-stage reversible watermarking.
>
> Put simply, WATERMARKLAB is a *framework*, while WAVES is a *benchmark*, analogous to PyTorch versus a model built on it. WATERMARKLAB provides foundational infrastructure. WAVES resembles a specific application on such a platform. Viewing our work as a mere extension of WAVES underestimates its systematic contribution. This distinction is precisely captured in our title: *WatermarkLab: **A Comprehensive Framework** for Robust Image Watermarks Benchmarking and Development*. We kindly ask you to reassess our work’s novelty and community impact accordingly.
>
> **2. Ease of Use**
>
> Our website provides a visualization interface and clear examples. Integrating a new method requires only implementing `embed` and `extract` (and optionally `recover`). The rest is handled automatically. For instance:
>
> ```python
> from numpy import ndarray
> import watermarklab as wl
> from typing import List, Any
> from watermarklab.utils.data import DataLoader
> from watermarklab.datasets import MS_COCO_2017_VAL_IMAGES
> from watermarklab.utils.basemodel import BaseWatermarkModel, Result
> from watermarklab.attackers.attackerloader import AttackersWithFactorsModel
>
>
> class YourWatermark(BaseWatermarkModel):
>     def __init__(self, bits_len: int, img_size: int, modelname: str):
>         super().__init__(bits_len, img_size, modelname)
>
>     def embed(self, cover_list: List[Any], secrets: List[Any]) -> Result:
>         pass  # Implement your embedding logic
>
>     def extract(self, stego_list: List[ndarray]) -> Result:
>         pass  # Implement your extraction logic
>
>     def recover(self, stego_list: List[ndarray]) -> Result:
>         pass  # Implement recovery if applicable
>
>
> # --- Setup Evaluation ---
> default_attackers = AttackersWithFactorsModel()  # Default attack suite
>
> your_payload = 100
> your_img_size = 400
> your_model = YourWatermark(bits_len=your_payload, img_size=your_img_size, modelname="YourModelName")
>
> # Load dataset (MS COCO 2017 validation set)
> dataset = MS_COCO_2017_VAL_IMAGES(im_size=your_img_size, bit_len=your_payload, image_num=500)
> dataloader = DataLoader(dataset, batch_size=64)
>
> # --- Run Evaluation ---
> wl.evaluate("save_results/PGWs", your_model, default_attackers, dataloader)
>
> # --- Visualize Results (Optional) ---
> wl.draw.plot_model_robustness_under_single_attack(
>     [f"save_results/PGWs/{your_model.modelname}/result_{your_model.modelname}.json"], "save_draw"
> )
> ```
>
> This design significantly lowers the barrier to entry for new researchers, facilitates reproducibility, and enables fair and efficient comparison across methods, thereby promoting standardization and collaboration within the watermarking community.
>
> **3. On Stable Diffusion v2.1-Base as a Weakness**
>
> We find it somewhat puzzling to consider the use of Stable Diffusion v2.1-Base as a major weakness. If the concern lies in the depth or breadth of our benchmark analysis, we are open to such feedback and willing to improve. However, citing the choice of a widely adopted and representative generative model, as a flaw seems unwarranted and does not reflect a substantive limitation of our work.
>
> Regardless of your final decision, we remain committed to maintaining WATERMARKLAB as a long-term open-source **PyPI** package to serve as reliable infrastructure for the watermarking community. Thank you again for your thoughtful review and constructive comments.

---

### Official Review · Reviewer_7B2h · 2025-10-29

**Soundness:** 3
**Presentation:** 2
**Contribution:** 3
**Rating:** 4
**Confidence:** 3

**Summary:**

This paper propose a new framework for watermark benchmarking. It covers 10 watermarking methods, including post-generation watermarking (PGW) and in-generation watermarking (IGW), and 34 attackers. The system integrates evaluation metrics, datasets, and visualization tools, enabling reproducible and fair comparisons.

**Strengths:**

1. The framework follows the previous works and includes abundant of watermarking and attacking methods.
2. The framework is extendable for the new watermarking method benchmarking in the future, which could be a key contribution for the research community of image watermarking.

**Weaknesses:**

1. Although the paper shows abundant of experimental results, it lacks of clear takeaway messages. All observations for different watermarks are stacked in several paragraphs, which is hard for the readers to get the key information. Moreover, high-level takeaway messages will be more inspiring to the researchers compared to the plain description of benchmarking results.
2. The necessary explanation for RQ-AUC are missed in the main body. Without any explanation, the readers will be confused with this new metric when they start to read the section about evaluation but have not reached the appendix.
3. The authors should explain why they choose PSNR as the quality metric in RQ-AUC. Using other quality metric such as FID or SSIM, will the overall ranking shown in Figure 8 change?
4. The authors highlighted the fair comparison as one of the main contribution of the proposed framework. But the analysis and explanation about the fairness of the comparison is missed in the paper. I think the discussion about how to design a fair comparison will be very insightful for the community.
5. The paper distinguished robust reversible watermarks such as CRMark with other PGWs. But based on my understanding it is still a special type of PGW. I cannot find any experimental results about this special watermarking. Since the paper highlight this special category of watermarking, the relevant experiments and analysis may be necessary to enable readers to recognize its value.

I am glad to adjust my score if the authors can address these questions.

**Questions:**

1. Typo: The index of last section "conclusion" is missed.
2. Why did you regard DctDwt as one of the"10 representative watermarking methods" but ignore this method in the evaluation (where only 9 methods are shown)?
3. The font size in Figure 7 and 8 is too small to read, especially for the watermark names on x-axis.
4. For each type of attacking, could there be a "sweet point" of strength balancing image quality and the attacking performance? For some strengths degrading the image quality a lot or not successfully attacking the watermark, it does not make sense to apply them in the real world application. But theses strengths seems to be also included in RQ-AUC. So maybe RQ-AUC is not able to show the attacking/robust performance at such a "sweet point" of attacking strength?

---

### Official Review · Reviewer_ay64 · 2025-10-30

**Soundness:** 3
**Presentation:** 2
**Contribution:** 2
**Rating:** 4
**Confidence:** 2

**Summary:**

This paper presents watermarklab, a comprehensive benchmarking framework for robust image watermarks. It supports all types of blind image watermarking algorithms. For benchmarking, it incorporates six categories of attack methods. And to demonstrate the results, in total nine watermarking schemes are tested in this paper to demonsrate the challenge in the proposed benchmark. This will be a good tool for helping the community evaluating and developing new image watermarking methods.

**Strengths:**

1.[comprehensiveness] The proposed method covers a wide range of image watermarking algorithms and attack methods. Lots of state-of-the-art watermarking methods are benchmarked in this paper as well.
2.As discussed in the paper, the benchmark is challenging, and I agree it can help the community develop better image watermarking methods that are more robust.
3.[related work] This paper presents a very comprehensive overview of related works. As a benchmark paper, this is necessary.

**Weaknesses:**

1.[typesetting] Please use "\citep" and "\citet" instead of the plain "\cite". The text in this draft is totally messed up wherever there is a citation. This could have been avoided if the authors had carefully read the draft.

2.[design] In real world internet, images are usually lossy-compressed in different extents. For instance, jpeg quality 70% may affect some non-robust watermarking methods. Does the proposed benchmark involve the consideration on how the images in real world are transmitted?

3.[new knowledge] This is a hard work -- I appreciate it. But I'm not seeing enough new knowledge from the huge amount of experimental observations. Namely, what do we learn from the comprehensive benchmark results? The knowledge is not something like "XXX method works best in our benchmark", but something like "XXX design pattern shows consistent robustness across different methods", or "XXX design pattern has some potential problem". Take one step further, this paper could have contributed much more to the community, than a plain observation, tool, and numbers.

**Questions:**

Please see the weaknesses section.

---

### Meta-Review · Area_Chair_dmZT · 2026-01-05

**Summary:**

In this work, the authors proposed WatermarkLab, a comprehensive framework for systematic benchmarking of robust image watermarks and the development of new methods.
However, the authors only rebutted to one reviewer's comments.
The major weaknesses of this work include:
1. One review appreciated that this is a hard work. But (s)he did not see enough new knowledge from the huge amount of experimental observations.
2. This paper shows abundant of experimental results but it lacks of clear takeaway messages.
3. The analysis and explanation about the fairness of the comparison is missed in the paper.
4. The contributions of this work are not clear and its difference from WAVES (ICML'24) is also not clear!

Since the authors did not further rebut and clarify the reviewers' concerns, this work is suggested to be rejected.

**Reviewer Scores:**

none

---

### Decision · Program_Chairs · 2026-01-26

Reject